# Spectral-Aware Reservoir Computing
# for Fast and Accurate Time Series Classification

**Shikang Liu** [1]  **Chuyang Wei** [1]  **Xiren Zhou** [1]  **Huanhuan Chen** [1]

## Abstract

Analyzing inherent temporal dynamics is a critical pathway for time series classification, where Reservoir Computing (RC) exhibits effectiveness and high efficiency. However, typical RC considers recursive updates from adjacent states, struggling with long-term dependencies. In response, this paper proposes a Spectral-Aware Reservoir Computing framework (SARC), incorporating spectral insights to enhance long-term dependency modeling. Prominent frequencies are initially extracted to reveal explicit or implicit cyclical patterns. For each prominent frequency, SARC further integrates a Frequency-informed Reservoir Network (FreqRes) to adequately capture both sequential and cyclical dynamics, thereby deriving effective dynamic features. Synthesizing these features across various frequencies, SARC offers a multi-scale analysis of temporal dynamics and improves the modeling of long-term dependencies. Experiments on public datasets demonstrate that SARC achieves state-of-the-art results, while maintaining high efficiency compared to existing methods.

## 1. Introduction

Time series classification (TSC) plays a significant role in numerous applications, such as diagnosing medical conditions (Malik et al., 2022), identifying financial fraud (Alghofaili et al., 2020), detecting anomalies in industrial processes (Wang et al., 2024), and classifying music genres (Chen et al., 2024b). This broad applicability has sparked considerable research interest in developing efficient and robust TSC methods (Iwana & Uchida, 2021).

[1]School of Computer Science and Technology, University of Science and Technology of China, Hefei, Anhui, China. Correspondence to: Xiren Zhou <zhou0612@ustc.edu.cn>, Huanhuan Chen <hchen@ustc.edu.cn>.

*Proceedings of the 42$^{nd}$ International Conference on Machine Learning*, Vancouver, Canada. PMLR 267, 2025. Copyright 2025 by the author(s).

Unlike other forms of sequential data, time series data typically record only a few numerical values at each time step, leading to insufficient information at individual time points (Wu et al., 2021). Consequently, researchers have focused on inherent temporal dynamics of time series for classification (Wang et al., 2022). This stems from the observation that each point in a time series is influenced by both adjacent point and several past ones, as real-world data often possesses cyclical patterns at multiple frequency scales (Wen et al., 2021). For example, weather data exhibits daily and seasonal cycles, while electrocardiograms display short-term repeats and circadian rhythms. Identifying such explicit or implicit cyclic patterns through spectral analysis and mining for prominent frequencies enhances the modeling and classification of time series data (Yi et al., 2024; Wu et al., 2023).

Following the widespread use of Deep Learning (DL), deep neural networks, particularly Convolutional Neural Networks (CNNs) and Recurrent Neural Networks (RNNs), have been applied to TSC tasks (Ismail Fawaz et al., 2019). CNN-based methods mainly slide convolutional kernels along the temporal dimension to capture changing patterns in time series (Franceschi et al., 2019; Tang et al., 2022). However, the locality of CNN kernels limits their ability to handle long-term dependencies. RNNs generate hidden states by combining the current input with past states (Shen et al., 2020). Nevertheless, due to issues like gradient vanishing or exploding, RNNs may struggle with memory decay, thereby hindering the capture of long-term dependencies. Recent studies (Wu et al., 2023; Yang et al., 2024) have also introduced spectral information into DL to enhance time series modeling. Yet, such methods remain resource-intensive and time-consuming, referring to the reliance on gradient descent for optimization.

As a subset of RNNs, Reservoir Computing (RC) demonstrates effectiveness and high efficiency in processing time series data (Chen et al., 2015). Typically, RC uses fixed, randomly initialized recurrent cells (the reservoir) to produce a rich set of high-dimensional states, facilitating the derivation of valuable dynamic features (Yan et al., 2024). A key characteristic of RC is that only the output layer is trainable, thus eliminating the need for backpropagation through time. Although such simplification significantly

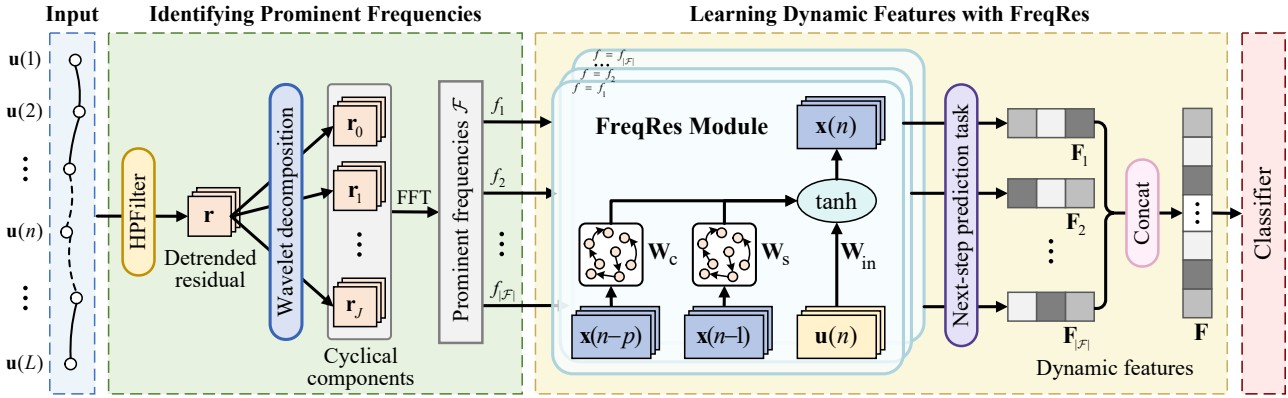

*Figure 1.* Overview of the SARC framework. Input series is decomposed into multi-scale cyclical components and analyzed spectrally to extract prominent frequencies (Section 4.1). An ensemble of FreqRes then captures temporal dynamics at local levels and across various cyclical intervals, deriving effective dynamic features for classification (Section 4.2).

reduces computational costs, a single and standard reservoir falls short in modeling long-term dependencies and cyclical patterns due to its recursive updates from adjacent states (Liu et al., 2025). Moreover, directly applying gating mechanisms (e.g., LSTM or GRU) to the reservoir also yields unsatisfactory results, as evident in our experimental study.

Addressing the above issues, we propose a novel Spectral-Aware Reservoir Computing (SARC) framework, which introduces spectral insights into RC, comprehensively capturing temporal dynamics within time series. As illustrated in Figure 1, SARC consists of two stages: Extracting multi-scale prominent frequencies to reveal explicit or implicit cyclical patterns, and Capturing dynamic information within each cyclical pattern for dynamic feature derivation. The first stage begins with detrending, which removes a long-term trend and preserves the cyclical residual. Wavelet decomposition then separates the residual into multi-scale components, representing multiple cyclical patterns. These components are analyzed through Fast Fourier Transform (FFT) to select frequencies with the highest amplitudes. Afterward, to enrich analytical scales, we extend the selected frequencies to their sub-harmonics. These comprise a set of multi-scale prominent frequencies, allowing SARC to capture a broader range of cyclical dynamics.

Subsequently, we integrate a modular component, namely the Frequency-informed Reservoir Network (FreqRes), for each extracted frequency. Unlike standard RC models, FreqRes features two reservoirs in its hidden layer, each representing connections between adjacent time steps and across cycles at the given frequency. When fitting a time series, FreqRes considers both of the two aspects to update the current state, enabling it to simultaneously capture sequential dynamics and long-span cyclical dynamics[1]. Building upon

this, an autoregressive task is applied to derive a "readout model" for each FreqRes. This model maps past states to future inputs, reflecting the dependencies between time steps. Hence, it encapsulates the captured dynamic information and serves as a dynamic feature. Synthesizing these features across various frequencies, SARC offers a multi-scale representation of time series and improves the modeling of long-term dependencies. As a result, even a simple classifier could be used for effective classification.

Summarily, our main contributions are as follows[2]:

- SARC innovatively introduces spectral insights to RC, capturing temporal dynamics both within neighborhoods and across long-span cycles. This provides a multi-scale analysis for time series, effectively modeling long- and short-term dependencies.

- Our framework is highly lightweight, requiring no gradient backpropagation. FreqRes independently operates on each sequence and directly solves for dynamic features using ridge regression. These allow for exceptional efficiency compared to baselines.

- SARC adapts to different RC implementations. Experimental studies on public datasets validate this flexibility, and also demonstrate that SARC achieves consistently superior performance when equipped with various RC implementations.

## 2. Related Work

We briefly review state-of-the-art TSC methods, followed by an introduction to reservoir computing and its applications.

**State-of-the-art TSC methods** can be broadly divided into

---

[1]Henceforth, "cyclical dynamics" refers to the dynamic information within a specific cyclical pattern or at a certain frequency.

[2]Code is available at https://github.com/ZOF-pt/SARC.

four categories: dictionary-based, shapelet-based, ensemble, and deep learning methods. As a latest dictionary-based method, Hydra (Dempster et al., 2023) transforms time series using random convolutional kernels arranged into groups, counting the kernel with the maximum and minimum response to represent key patterns. In contrast, Shapelet Transform (ST; Hills et al. 2014) identifies the most discriminative subsequences, or shapelets, from time series, and then converts them into features for classification. Regarding ensemble methods, Collection of Transformation Ensembles (COTE; Lines et al. 2018) integrates multiple classifiers, such as Time Series Forest (Deng et al., 2013), Bag-of-SFA-Symbols (Schäfer, 2015), and Elastic Ensemble (Lines & Bagnall, 2015), to enhance TSC performance.

On the other hand, deep learning methods leverage neural networks to automatically extract features from time series data, achieving remarkable results in various TSC tasks (Ismail Fawaz et al., 2019). For example, InceptionTime (Ismail Fawaz et al., 2020) employs an ensemble of Inception modules to capture complex temporal patterns. LSTNet (Lai et al., 2018) combines CNNs and RNNs to extract local dependencies and incorporates a recurrent-skip layer to handle long-term dependencies. Another notable method is TimesNet (Wu et al., 2023), which is more related to our approach as it also introduces periodicity. TimesNet uses the periods obtained from Fourier Transform to segment and transform time series into 2D formats, then employs vision backbones for classification. Besides, transformers (Wu et al., 2021; Zhou et al., 2022) have also gained attention for their ability to handle long-term dependencies. Despite the effectiveness of these methods, they are all constrained by the high computational and time costs.

**Reservoir Computing** uses reservoirs with randomly interconnected neurons to adeptly capture temporal dynamics from time series (Yan et al., 2024). Unlike traditional neural networks, it offers an efficiency advantage by maintaining fixed connection weights. The foundational concepts of RC were introduced via two primary models: Echo State Networks (ESNs; Jaeger 2001) and Liquid State Machines (LSMs; Maass et al. 2002). Following these, researchers have proposed various alternatives such as the Echo State Gaussian Process (Chatzis & Demiris, 2011), Delayed Feedback Reservoir (Appeltant et al., 2011), DeepESN (Gallicchio et al., 2018), and 2D-ESN (Chen et al., 2024a). Recently, RC has been increasingly applied to time series classification, demonstrating its efficacy in many tasks (Liu et al., 2024). For instance, ConvMESN (Ma et al., 2019) incorporates reservoirs with varying skips to compute multi-timescale echo states, followed by a 1D convolutional layer for feature extraction. Additionally, rmESN (Bianchi et al., 2021) fits a linear transformation between adjacent echo states, using the weights as a representation of temporal dynamics. However, simplifying the reservoir in standard

RC models can sometimes limit their accuracy, suggesting a need for more complex or nuanced RC architectures.

## 3. Preliminaries

Our framework adapts to various RC implementations. For a clear illustration of its core improvements, we formalize the simplest form of RC, namely the ESN, which underpins the subsequent methodological explanation.

An ESN consists of three main components: fixed, randomly initialized linear input layer and reservoir, along with a readout layer. During iteration, its hidden states are updated based on the previous state and the current input. The readout layer then converts these states into final outputs. Formally, the iterative and output formulas are as follows:

$$\mathbf{x}(n) = f(\mathbf{W}\mathbf{x}(n-1) + \mathbf{W}_{\text{in}}\mathbf{u}(n)), \qquad (1)$$

$$\mathbf{y}(n) = \mathbf{W}_{\text{out}}\mathbf{x}(n), \qquad (2)$$

where $\mathbf{u}(n)$, $\mathbf{x}(n)$, and $\mathbf{y}(n)$ denote the input, hidden state, and output at time step $n$; $\mathbf{W}_{\text{in}}$ and $\mathbf{W}_{\text{out}}$ are the input and readout weight matrices; $\mathbf{W}$ represents the reservoir weight matrix; and $f$ is an activation function, typically $tanh$. According to Equation (1), a typical ESN only considers recursion between adjacent hidden states, struggling to capture long-term dependencies within time series.

## 4. Methodology

The proposed SARC framework comprises two main parts: 1) Extracting multi-scale prominent frequencies to identify potential cyclical patterns; 2) Capturing cyclical dynamics via FreqRes and deriving dynamic features. These parts are illustrated in Figure 1 and further elaborated below.

### 4.1. Identifying Potential Cyclical Patterns

To identify cyclical patterns and capture long-span cyclical dynamics, it is crucial to extract prominent frequencies from time series data. A straightforward approach involves selecting the top-$k$ amplitudes from FFT (Wu et al., 2023). However, this strategy risks neglecting higher frequencies (Duhamel & Vetterli, 1990; Zhou et al., 2022). Additionally, if the time series shows a clear non-cyclical trend, the highest amplitudes tend to concentrate near zero frequency, rendering the extraction less meaningful. In response, SARC detrends the time series and decomposes the residual into multi-scale cyclical components, after which the prominent frequencies are extracted from each scale.

#### 4.1.1. DETRENDING AND CYCLICAL DECOMPOSITION

Given a time series $\mathbf{u} = \{\mathbf{u}(n)\}_{n=1}^{L}$ of length $L$ with $C$ variables, detrending is initially conducted to reduce non-cyclical influences. To this end, we apply the Hodrick-

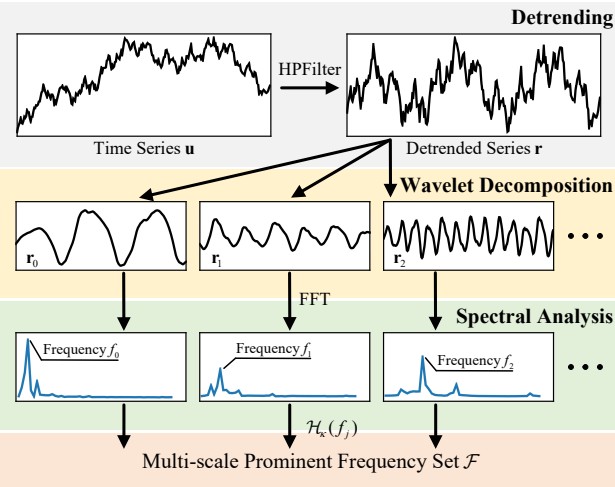

*Figure 2.* An illustration of prominent frequency extraction: The series **u** is first detrended and decomposed into cyclical components $\{\mathbf{r}_j\}_{j=0}^{J}$. Each component then undergoes FFT to extract the highest amplitude and the corresponding frequency $f_j$.

Prescott filter (Hodrick & Prescott, 1997) featuring high efficiency and effectiveness in estimating long-term trends from time series data (Wen et al., 2021). This involves minimizing the following objective function:

$$\hat{\boldsymbol{\tau}}(n) = \arg\min_{\boldsymbol{\tau}} \left\{ \frac{1}{2} \sum_{n=1}^{N} (\mathbf{u}(n) - \boldsymbol{\tau}(n))^2 \right.$$
$$\left. + \lambda \sum_{n=2}^{N-1} [(\boldsymbol{\tau}(n+1) - 2\boldsymbol{\tau}(n) + \boldsymbol{\tau}(n-1)]^2 \right\}, \quad (3)$$

where $\hat{\boldsymbol{\tau}} = \{\hat{\boldsymbol{\tau}}(n)\}_{n=1}^{L}$ is the estimated trend of **u**; $\lambda$ is the smoothing parameter: a smaller $\lambda$ retains more short-term fluctuations in $\hat{\boldsymbol{\tau}}$, while a larger $\lambda$ makes it smoother. Typically, $\lambda$ is set to 1600, and this problem can be solved using dynamic programming.

Through the above, the long-term trend is removed from the original time series, yielding a cyclical residual $\mathbf{r} = \mathbf{u} - \hat{\boldsymbol{\tau}}$. This residual represents a mixture of multiple cyclical influences within the data. Decoupling the mixture, we employ wavelet decomposition and separate $\mathbf{r}$ into $J + 1$ cyclical components at different scales, including one approximation component $\mathbf{r}_a$ and $J$ levels of detail components $\mathbf{r}_{d,j}$, $j \in \{1, 2, \ldots, J\}$. Each of these components reflects a potential cyclical pattern, aligned with a prominent frequency to be extracted. Note that, for simplicity and uniformity in notation, we will henceforth adopt the following conventions: $\mathbf{r}_0 := \mathbf{r}_a$, and $\mathbf{r}_j := \mathbf{r}_{d,j}$.

### 4.1.2. PROMINENT FREQUENCY EXTRACTION

Once wavelet decomposition is complete, we extract the corresponding prominent frequencies. Each cyclical compo-

nent is analyzed through FFT:

$$\mathbf{A}_j = |\text{FFT}(\mathbf{r}_j)|, \quad f_j = \underset{f \in \{1, \ldots, \lfloor \frac{L}{2} \rfloor\}}{\arg\max} (\mathbf{A}_j). \quad (4)$$

Specifically, $|\text{FFT}(\cdot)|$ calculates the magnitude of the FFT results. $\mathbf{A}_j$ contains the non-normalized amplitudes of $\mathbf{r}_j$ at different frequencies. The most prominent frequency $f_j$ is determined by selecting the highest amplitude in $\mathbf{A}_j$. This frequency accurately reflects the cyclical pattern, with a cycle length of $p_j = [L/f_j]$, as it is free from long-term trends and other prominent frequencies.

To fully capture the cyclical dynamics at these frequencies, we design to model the multi-scale dependencies spanning multiple cycles. Concretely, the selected frequencies are extended to their sub-harmonics, comprising a set of multi-scale prominent frequencies $\mathcal{F}$:

$$\mathcal{F} = \bigcup_{j=0}^{J} \mathcal{H}_\kappa(f_j) = \bigcup_{j=0}^{J} \left\{ \frac{f_j}{k} \mid k \in \mathbb{N}^+, k p_j < \kappa \right\}. \quad (5)$$

Here, $\mathcal{H}_\kappa$ denotes the transformation from a frequency to its sub-harmonics with a threshold parameter $\kappa$. Each sub-harmonic $f_j/k$ characterizes a dependency spanning $k$ cycles of length $p_j$, constrained by $\kappa$. Therefore, $\kappa$ defines the maximum span of dependencies that needs to be modeled. Typically, a larger $\kappa$ implies more meticulous analysis but also leads to higher computational complexity.

### 4.2. Frequency-informed Reservoir Network (FreqRes)

For each frequency $f \in \mathcal{F}$, SARC integrates a FreqRes module. Unlike a standard reservoir with only adjacent connections, FreqRes features an additional cycle connection, which updates the hidden state by combining states from both previous steps and cycles. This improvement simultaneously captures the sequential dynamics and long-span cyclical dynamics for more comprehensive modeling of time series. Moreover, given the diversity of cycle lengths, integrating multiple FreqRes modules facilitates a multi-scale analysis of the temporal dynamics, thus effectively handling both short- and long-term dependencies. Incorporating such a cycle connection into any existing RC model results in a different FreqRes implementation. For clarity, we now propose the iteration of FreqRes and the derivation of dynamic features based on an ESN model introduced in Section 3.

#### 4.2.1. ITERATION OF FREQRES

As shown in Figure 3, FreqRes iterates in a 2D grid-like pattern. Given an input time series $\mathbf{u} \in \mathbb{R}^{L \times C}$ and a specific frequency $f$, the process starts with an initial state $\mathbf{x}(0) = \mathbf{0}$. Step by step, the raw inputs are weighted through a linear input layer and fed into a hidden layer containing two independent reservoirs. Each reservoir is composed

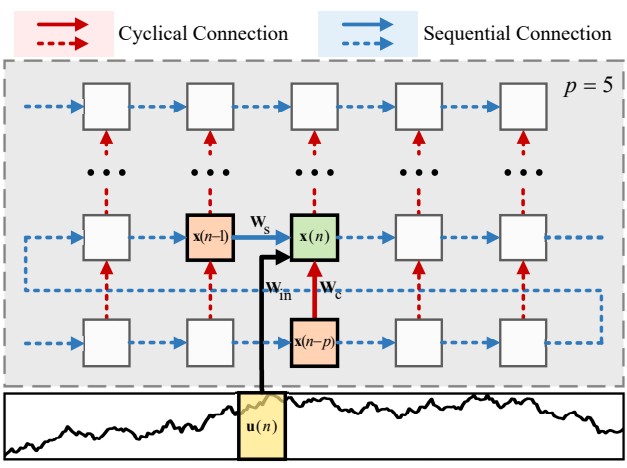

Cyclical Connection    Sequential Connection

*Figure 3.* An example of the FreqRes iteration with a cycle $p = 5$. Hidden states are arranged in a 2D layout, each row containing $p$ contiguous time steps. Hidden state update considers the state from the previous time step (left) and cycle (below). $\mathbf{W}_s$ and $\mathbf{W}_c$ represent the influence of the two independent reservoirs.

**Algorithm 1** Spectral-Aware Reservoir Computing

**Input:** Time series $\mathbf{u} \in \mathrm{R}^{L \times C}$, smoothing parameter $\lambda$, threshold parameter $\kappa$, regularization parameter $\zeta$.
**Initialize:** $\mathcal{F} \leftarrow \emptyset$, $\mathbf{x}(0) \leftarrow \mathbf{0}$, $\mathbf{W}_{in}, \mathbf{W}_s, \mathbf{W}_c \sim \mathcal{N}(0, 1)^3$.
**Output:** Dynamic feature $\mathbf{F}$.
1: $\hat{\boldsymbol{\tau}} \leftarrow \text{HPFilter}(\mathbf{u}, \lambda)$, $\mathbf{r} \leftarrow \mathbf{u} - \hat{\boldsymbol{\tau}}$.          // Detrending.
2: $\{\mathbf{r}_j\}_{j=0}^J \leftarrow \text{WaveDec}(\mathbf{r})$.          // Wavelet decomposition.
3: **for** $j = 0$ to $J$ **do**
4:     $\mathbf{A}_j \leftarrow |\text{FFT}(\mathbf{r}_j)|$ .
5:     $f_j \leftarrow \underset{f \in \{1, \ldots, \lfloor \frac{L}{2} \rfloor\}}{\arg\max} (\mathbf{A}_j)$.
6:     $\mathcal{F} \leftarrow \mathcal{F} \cup \{\frac{f_j}{k} \mid k \in \mathbb{N}^+, kp_j < \kappa\}$.
7: **end for**          // Extract prominent frequencies.
8: **for** $f \in \mathcal{F}$ **do**
9:     $p \leftarrow \lceil \frac{L}{f} \rceil$.
10:     **for** $n = 1$ to $L$ **do**
11:         Update state $\mathbf{x}(n)$ according to Eq. (1) and (6).
12:         $\tilde{\mathbf{x}}(n) \leftarrow [\mathbf{x}(n-1); \mathbf{x}(n-p)]$.
13:     **end for**
14:     $\mathbf{U}, \mathbf{X} \leftarrow \mathbf{u}[p : L+1], \tilde{\mathbf{x}}[p : L+1]$.
15:     $\mathbf{W}_{out} \leftarrow \mathbf{U}\mathbf{X}^T(\mathbf{X}\mathbf{X}^T + \zeta \mathbf{I})^{-1}$.          // Ridge regression.
16:     $\mathbf{F}_f \leftarrow \text{flatten}(\mathbf{W}_{out})$.
17: **end for**          // Derive dynamic features from $p$.
18: $\mathbf{F} = \text{concatenate}(\mathbf{F}_f \text{ for all } f \in \mathcal{F})$.

of randomly connected neurons, reflecting the influence of historical data on present states. One of the reservoirs accounts for variations between adjacent time steps, while the other focuses on dynamics across cycles with a span of $p = \lceil L/f \rceil$. During the first $p$ steps ($n \le p$), hidden states of the previous cycle are not available; hence, FreqRes adheres to the base iteration formula in Equation (1). Then, for $n > p$, the hidden states are updated as follows:

$$\mathbf{x}(n) = f(\mathbf{W}_s\mathbf{x}(n-1) + \mathbf{W}_c\mathbf{x}(n-p) + \mathbf{W}_{in}\mathbf{u}(n)). \quad (6)$$

Here, $\mathbf{W}_{in}$ is the input weight matrix. $\mathbf{W}_s$ and $\mathbf{W}_c$ are the reservoir weight matrices that describe the interconnections among neurons within each reservoir, modulating the influence of states from the previous time step and cycle. Similar to the original ESN, these three matrices are randomly initialized, remaining fixed and untrainable.

In alignment with the added cyclical connection, the output formula of FreqRes has been refined accordingly. Rather than mapping $\mathbf{x}(n)$ directly, the output layer now uses a combined state $\tilde{\mathbf{x}}(n) = [\mathbf{x}(n-1); \mathbf{x}(n-p)]$ to determine the target output $\mathbf{y}(n)$:

$$\mathbf{y}(n) = \mathbf{W}_{out}\tilde{\mathbf{x}}(n) = \begin{bmatrix} \mathbf{W}_{out,s} & \mathbf{W}_{out,c} \end{bmatrix} \begin{bmatrix} \mathbf{x}(n-1) \\ \mathbf{x}(n-p) \end{bmatrix}. \quad (7)$$

Here, $\mathbf{W}_{out}$ is partitioned into $\mathbf{W}_{out,s}$ and $\mathbf{W}_{out,c}$ to map the contributions from $\mathbf{x}(n-1)$ and $\mathbf{x}(n-p)$, respectively.

#### 4.2.2. DERIVATION OF DYNAMIC FEATURES

Leveraging the rich hidden states from each FreqRes, we derive a representative "readout model" to serve as the dynamic features. Concretely, an autoregressive "next-step

prediction task" is implemented. It requires the output $\mathbf{y}(n)$ to approximate input $\mathbf{u}(n)$, enabling the output layer to carry the transformation from previous states (both the previous time step and cycle) to current sequence values. This approach models the recursive relationships within the input time series. Consequently, the derived readout model, $\mathbf{y} = \mathbf{W}_{out}\tilde{\mathbf{x}}$, is considered to encapsulate the temporal dynamics across both sequential and cyclical dimensions, and thus serve as the dynamic features from frequency $f$. In practice, this readout model is represented by $\mathbf{W}_{out}$, which can be solved using ridge regression:

$$\mathbf{W}_{out} = \mathbf{U}\mathbf{X}^T(\mathbf{X}\mathbf{X}^T + \zeta \mathbf{I})^{-1}, \quad (8)$$

where $\mathbf{U} = [\mathbf{u}(p), \mathbf{u}(p+1), ..., \mathbf{u}(L)]$ is the target matrix; $\mathbf{X} = [\tilde{\mathbf{x}}(p), \tilde{\mathbf{x}}(p+1), ..., \tilde{\mathbf{x}}(L)]$ is the combined state matrix; $\zeta$ is a regularization parameter; and $\mathbf{I}$ is the identity matrix. By flattening and concatenating the $\mathbf{W}_{out}$ derived from all frequencies, final dynamic features $\mathbf{F}$ are obtained. For further insights into the overall procedure, we detail our framework in Algorithm 1.

## 5. Experiments

In this section, we evaluate the SARC on public benchmarks. As aforementioned, the FreqRes is not confined to a specific RC model. By default, we implement FreqRes based on a Bidirectional ESN (BiESN), which is demonstrated to be optimal across four conventional RC models in our ablation

---

[3]For brevity, input weight scaling for $\mathbf{W}_{in}$ and spectral radius adjustments for $\mathbf{W}_s$ and $\mathbf{W}_c$ are omitted in Algorithm 1.

*Table 1.* Comparison of accuracy and F1-score between the proposed SARC and baselines across 128 datasets. Bold values represent the maximum scores for each metric. P-values > 0.05 are underlined, indicating no statistically significant difference. "Wins/Ties/Losses" count the datasets where each baseline performed better, equally, or worse than SARC, respectively.

| Methods | Accuracy | | | | F1-score | | | |
|---|---|---|---|---|---|---|---|---|
| | Avg. Acc. | Avg. Rank | P-value | Wins/Ties/Losses | Avg. F1 | Avg. Rank | P-value | Wins/Ties/Losses |
| rmESN | 0.7689 | 6.7266 | 1.60E-20 | 9/5/114 | 0.7560 | 6.7265 | 7.08E-20 | 7/5/116 |
| ConvMESN | 0.8108 | 5.8164 | 4.01E-16 | 20/3/105 | 0.7929 | 5.9487 | 3.15E-15 | 18/2/108 |
| TimesNet | 0.6867 | 8.3984 | 1.76E-22 | 1/1/126 | 0.6714 | 8.1624 | 8.41E-23 | 2/3/123 |
| InceptionTime | 0.8352 | 4.5234 | 1.60E-03 | 42/10/76 | 0.8313 | 4.4188 | 4.03E-03 | 45/6/77 |
| COTE | 0.8484 | 3.7539 | 4.03E-01 | 59/8/61 | 0.8405 | 3.9017 | 5.48E-01 | 56/8/64 |
| Hydra | 0.8423 | 4.2695 | 3.78E-03 | 41/10/77 | 0.8377 | 4.3120 | 7.52E-03 | 45/9/74 |
| Rocket | 0.8450 | 3.8008 | 2.21E-01 | 56/12/60 | 0.8410 | 3.7991 | 1.73E-01 | 53/14/61 |
| MiniRocket | 0.8411 | 4.3711 | 1.81E-03 | 42/10/76 | 0.8368 | 4.3761 | 2.50E-03 | 44/10/74 |
| **SARC** | **0.8508** | **3.3398** | - | - | **0.8449** | **3.3547** | - | - |

studies (Section 5.3)[4]. All experiments are conducted using Python 3.11 on a desktop with an Intel Core i7-14700KF CPU, and an NVIDIA GeForce RTX 4090D GPU.

### 5.1. Experimental Settings

#### 5.1.1. DATASETS

We use the full UCR Time Series Archive (Dau et al., 2019) with 128 datasets spanning various applications such as activity recognition, health monitoring, and spectrum analysis. These datasets encompass time series lengths from 15 to 2844, classes ranging from 2 to 60, and training sets from 16 to 8926 samples. Such diverse characteristics and complexity enable a thorough evaluation of our method.

#### 5.1.2. COMPARATIVE METHODS

The proposed SARC is evaluated against 8 state-of-the-art methods, which have been introduced in the Related Work. These include two RC-based methods: **rmESN** (Bianchi et al., 2021) and **ConvMESN** (Ma et al., 2019); two deep neural networks: **InceptionTime** (Ismail Fawaz et al., 2020) and **TimesNet** (Wu et al., 2023); and four advanced methods on the UCR Archive, namely **COTE** (Lines et al., 2018), **Rocket** (Dempster et al., 2020), **MiniRocket** (Dempster et al., 2021), and **Hydra** (Dempster et al., 2023). Similar to our approach, TimesNet identifies cyclical patterns within time series for classification. InceptionTime and COTE both demonstrate strong performance. Additionally, Rocket, MiniRocket, and Hydra are relatively lightweight, making them suitable for assessing SARC's efficiency.

#### 5.1.3. IMPLEMENTATION DETAILS

In our experiments, all datasets are standardized to maintain uniformity across evaluations. For datasets with variable

---

[4] FreqRes based on BiESN involves: performing forward and backward iterations as in Eq. (6), concatenating the bidirectional hidden states and calculating dynamic features using Eq. (8).

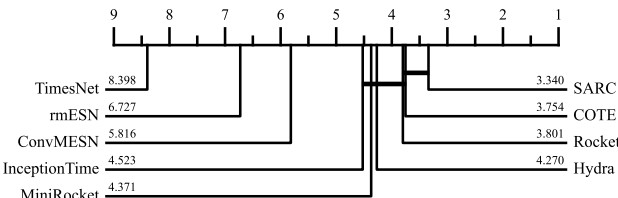

*Figure 4.* Critical Difference diagram for accuracy on 128 datasets.

lengths or missing values, we first interpolate middle NaNs (i.e., those surrounded by real values), then align real values to the right end of the tensor, and finally fill leading NaNs with zeros. Key hyperparameters are determined through a five-fold cross-validation on the training set, selecting input scaling from {0.5, 1, 2, 4}, spectral radii from {0.4, 0.6, 0.8}, regularization $\zeta$ from {0.5, 1}, and leaky rates ranging from 0 to 0.8 in 0.2 increments. The reservoir size is set to 10, the connectivity is 1, and the threshold $\kappa$ is set to 100. For classification, we concatenate the derived dynamic features with the max-pooled hidden states and feed them to a default Ridge classifier. The experiments are repeated five times using different random seeds. Regarding the baselines, we use official implementations or configurations from the original papers. More detailed information is provided in Section A.1 of the Appendix.

### 5.2. Classification Performance Evaluation

#### 5.2.1. ACCURACY AND F1 SCORE

We analyze accuracy and F1 scores across 128 datasets from multiple perspectives, including average metrics, rankings, the Wilcoxon Signed-Rank Test, and counts of wins/ties/losses. From the results in Table 1, SARC achieves the overall best performance, surpassing state-of-the-art methods like COTE, Rocket, and Hydra. It not only boasts the highest average accuracy of 0.8508 and the highest average F1 score of 0.8449, but also maintains the top average

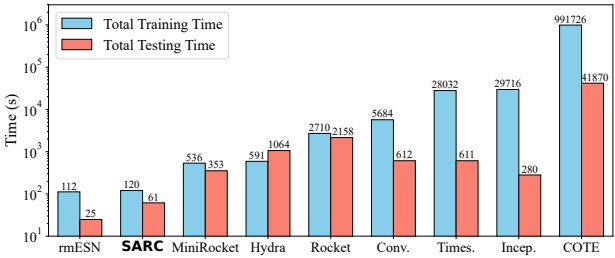

*Figure 5.* Runtime comparison (in log scale) on 128 UCR datasets.

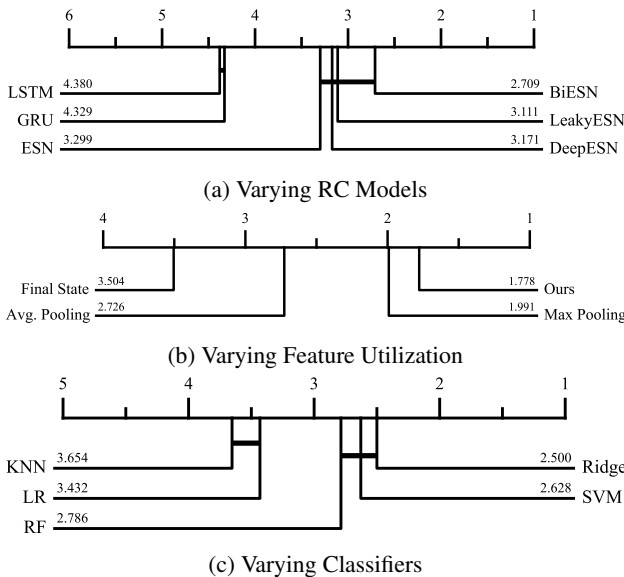

*Figure 6.* Critical Difference diagrams for ablation study on 128 datasets, based on accuracy rankings.

ranking. Statistically, SARC exhibits significant differences from most comparative methods and consistently wins more in pairwise comparisons. Compared to rmESN and ConvMESN, which are also RC-based, SARC demonstrates significant performance advantages, with average accuracy improvements of 8.19% and 4.00%, respectively, highlighting the benefits of incorporating spectral awareness.

### 5.2.2. COMPLEXITY AND TIME

The main factors affecting SARC's complexity include sample size $M$, series length $L$, threshold $\kappa$, and reservoir size $S$. According to experiments, SARC's runtime is primarily spent on iteration of FreqRes modules and computation of dynamic features, with a complexity of $O(\kappa M(S^2 L + S^3))$. This means that under specific settings, the runtime is nearly linear with the data complexity (characterized by $M$ and $L$), showcasing exceptional efficiency.

Figure 5 compares the runtime of various methods on 128 datasets. Clearly, SARC is faster than most baseline methods. Compared to ConvMESN and InceptionTime, which require gradient descent, SARC is tens to hundreds of times faster in training. Considering relatively lightweight approaches like MiniRocket, SARC's training is still several times faster, and the advantage is even more pronounced in the testing phase. This exceptional efficiency stems from the absence of gradient backpropagation and the high parallelizability among FreqRes modules. The only method that slightly exceeds SARC in runtime is rmESN; however, its accuracy lags substantially, rendering it less comparable.

### 5.3. Ablation Study

#### 5.3.1. IMPACT OF RC MODELS

We evaluate SARC based on four conventional RC models: **ESN**, **LeakyESN**, **BiESN**, and **DeepESN**, and two gated RNN models: **LSTM** and **GRU**, with implementation details provided in **Section A.2** of the **Appendix**. From Figure 6a and Table 2, we find no significant differences when using the four RC models, while LSTM and GRU lag behind. Compared to BiESN, other RC models show only minor declines, with a maximum reduction of 1.55% in accuracy

and 1.61% in F1 score. Even using the weakest ESN, SARC remains competitive with the state-of-the-art InceptionTime, highlighting its superiority and flexibility. This demonstrates SARC's potential to accommodate advancements in the field of RC and leverage its strengths. On the other hand, a notable accuracy decline is observed when implementing SARC based on LSTM and GRU, which confirms that directly applying gating mechanisms in RC without adequate training often leads to unsatisfactory results.

#### 5.3.2. IMPACT OF FEATURE UTILIZATION

To validate the performance improvement from incorporating the derived dynamic features, we compare the utilized features with three commonly used RNN feature types: **Max Pooling** and **Average Pooling** of hidden states, and the **Final State**. From the results in Figure 6b and Table 2, using only Max Pooling shows a modest performance decline, while the others lead to significant decreases in average accuracy by 5.97% and 10.81%. This is because the derived dynamic feature effectively integrates past and present information, encapsulating the inherent temporal dynamics, which leads to improved performance.

#### 5.3.3. IMPACT OF CLASSIFIERS

We also assess the impact of different classifiers, including Logistic Regression (**LR**), Random Forest (**RF**), K-Nearest Neighbors (**KNN**), Support Vector Machines (**SVM**), and Ridge Regression (**Ridge**) classifier. Figure 6c shows that Ridge achieves the best performance, which is why we use it by default. RF and SVM are relatively weaker, yet not statis-

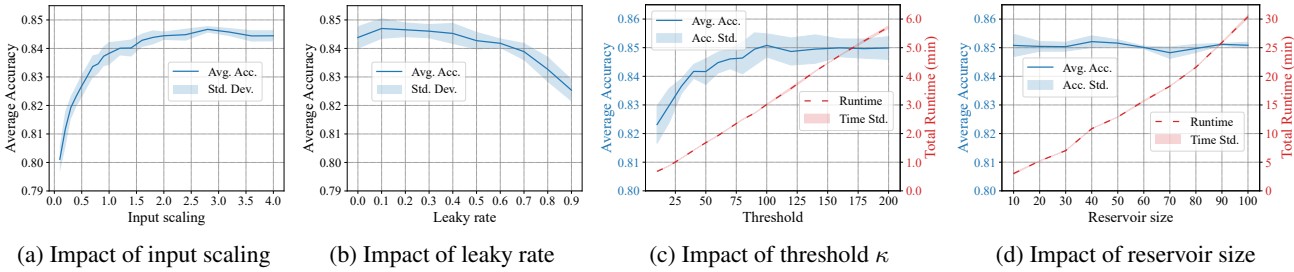

(a) Impact of input scaling     (b) Impact of leaky rate     (c) Impact of threshold $\kappa$     (d) Impact of reservoir size

*Figure 7.* Impact of input scaling, leaky rate, threshold $\kappa$ and reservoir size on average accuracy and total runtime across 128 datasets.

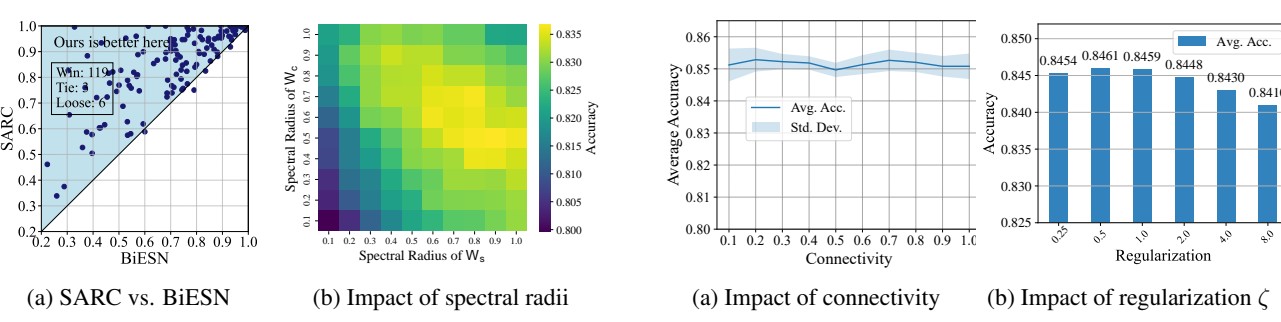

(a) SARC vs. BiESN    (b) Impact of spectral radii     (a) Impact of connectivity    (b) Impact of regularization $\zeta$

*Figure 8.* (a) Accuracy comparison between SARC and BiESN, and (b) the impact of spectral radius on average accuracy.

*Figure 9.* Impact of varying reservoir connectivity and regularization on average accuracy across 128 datasets.

tically significant, indicating that the derived features adapt well to various classifiers. However, both LR and KNN yield inferior outcomes. Results in Table 2 also confirm this lag, suggesting their difficulties in integrating features across multi-scale prominent frequencies.

### 5.3.4. COMPARISON WITH BiESN

To further investigate the contribution of spectral awareness in the SARC framework, we compare it with a variant lacking this mechanism, namely a single BiESN. As illustrated in Figure 8a, SARC consistently surpasses the BiESN in accuracy across 119 datasets, achieving over a 10% improvement in 65 cases. This highlights that incorporating spectral insights into RC models enhances the capture of temporal dynamics within time series data, leading to more discriminative dynamic features for classification.

### 5.4. Hyperparameter Study

We now turn to the impact of key hyperparameters in SARC. Figure 7a shows a notable rise in accuracy as input scaling increases to about 2.8, beyond which the changes become minimal, suggesting that a larger input scaling might be preferable. In contrast, Figure 7b indicates that average accuracy slightly improves with a leaky rate increase from 0 to 0.2, but diminishes with further increases, recommending a smaller leaky rate close to zero.

Figure 7c also shows that average accuracy markedly im-

proves as threshold $\kappa$ increases. This confirms that extending prominent frequencies provides richer scale insights and effectively enhances TSC performance. Unexpectedly, reservoir size has minimal impact on accuracy but significantly elevates runtimes, as depicted in Figure 7d. Thus, we have fixed it at 10 for efficiency. From Figure 8b, Spectral radii markedly influence SARC's performance, with optimal accuracy near a spectral radius of 0.8 for $\mathbf{W}_s$ and 0.6 for $\mathbf{W}_c$. This implies a weighting between sequential and cyclical dynamics: the former should slightly outweigh the latter.

From Figure 9a, accuracy remains stable as connectivity increases, with a dip around 0.5 and peaks near 0.1 or 1.0. Conversely, the standard deviation is lowest at 0.5, and highest at 0.1, highlighting a trade-off between accuracy and stability. Even so, the standard deviation stays below 0.5%, demonstrating the overall stability of SARC. Figure 9b further illustrates that average accuracy is highest around $\zeta = 0.5$ to 1.0. Low $\zeta$ values can cause overfitting, leading to the capture of excessive details and reducing generalizability. High $\zeta$ values may result in underfitting, as the regularization suppresses informative temporal dynamics, diminishing the category-discriminability of the derived features.

## 6. Conclusion

This paper proposes SARC, a novel Reservoir Computing framework with spectral awareness, integrating multiple FreqRes modules corresponding to multi-scale prominent

frequencies. These modules simultaneously capture both sequential and cyclical dynamic information within time series, providing a comprehensive analysis of temporal dynamics and deriving effective dynamic features for TSC. Experiments demonstrate that SARC achieves state-of-the-art accuracy while maintaining exceptional efficiency. Future work could delve into the mechanisms by which spectral information influences TSC performance or explore applications in other tasks such as time series anomaly detection.

## Acknowledgments

This work was supported in part by the National Natural Science Foundation of China (No. 62206261, 62137002, 62176245), in part by the Fundamental Research Funds for the Central Universities (No. WK2150110039).

## Impact Statement

This paper presents work whose goal is to advance the field of Machine Learning. There are many potential societal consequences of our work, none of which we feel must be specifically highlighted here.

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

# A. Implementation Details

## A.1. Comparative Methods

The eight advanced baselines are sourced from official implementations to ensure reproducibility and consistency. Specifically, **TimesNet** are from the `TSLib` package[5]. **InceptionTime**, **COTE**, **Rocket**, and **MiniRocket** are implemented using the `sktime` package[6]. Additionally, **ConvMESN** and **Hydra** are available from their respective repositories[78], while **rmESN** follows the hyperparameter settings from its original paper.

## A.2. SARC Variants

The variants of SARC in Table 2, achieved by employing different RC models or RNNs, are detailed as follows:

- **SARC with ESN**: Iterations and feature derivation of FreqRes modules conform to equations (6)-(8). The leaky rate is fixed at 0. Other key hyperparameters not mentioned here follow the settings in Section 5.1.

- **SARC with LeakyESN**: A leaky integrator is incorporated into equation (6). The update of $\mathbf{x}(n)$ includes leakage from $\mathbf{x}(n-1)$, weighted by the leaky rate. The leaky rate for all FreqRes is uniformly set and chosen from $\{0, 0.2, 0.4, 0.6, 0.8\}$.

- **SARC with DeepESN**: For a given layer, the hidden states derived from equation (6) serve as inputs for the next layer, and those from the final layer are used for subsequent processing. The number of layers is selected from $\{2, 3\}$, and the leakage is also adopted.

- **SARC with LSTM / GRU**: For each extracted frequency, a single-layer LSTM / GRU is implemented, with its hidden dimension set to 10, matching the reservoir size of the default SARC.

# B. Complexity Analysis

In the SARC workflow, operations such as the HP filter, wavelet decomposition, FFT, and Ridge Classifier are efficiently handled by well-established libraries, ensuring rapid computation. The runtime is mainly spent on the iteration of FreqRes modules and dynamic feature derivation. We demonstrate that the complexity of these processes is $O(\kappa M(S^2 L + S^3))$, where $M$ denotes the sample size, $L$ the series length, $\kappa$ the period limit, and $S$ the reservoir size.

---

[5]https://github.com/thuml/
Time-Series-Library
[6]https://github.com/sktime/sktime
[7]https://github.com/qianlima-lab/ConvMESN
[8]https://github.com/angus924/hydra

*Table 2.* Ablation results on 128 UCR datasets.

| | Avg. Acc. | Avg. F1-score |
|---|---|---|
| *Varying RC Models* | | |
| **BiESN** | **0.8508** | **0.8449** |
| $\rightarrow$ ESN | 0.8353 (-1.55%) | 0.8288 (-1.61%) |
| $\rightarrow$ LeakyESN | 0.8416 (-0.92%) | 0.8350 (-0.99%) |
| $\rightarrow$ DeepESN | 0.8394 (-1.14%) | 0.8325 (-1.24%) |
| $\rightarrow$ LSTM | 0.7896 (-6.12%) | 0.7852 (-5.97%) |
| $\rightarrow$ GRU | 0.7926 (-5.82%) | 0.7874 (-5.75%) |
| *Varying Feature Utilization* | | |
| **Utilized Feature** | | |
| $\rightarrow$ Max Pooling | 0.8365 (-1.43%) | 0.8301 (-1.48%) |
| $\rightarrow$ Avg. Pooling | 0.7911 (-5.97%) | 0.7835 (-6.14%) |
| $\rightarrow$ Final Staste | 0.7427 (-10.81%) | 0.7343 (-11.06%) |
| *Varying Classifiers* | | |
| **Ridge** | | |
| $\rightarrow$ LR | 0.8253 (-2.55%) | 0.8069 (-3.80%) |
| $\rightarrow$ RF | 0.8391 (-1.17%) | 0.8309 (-1.40%) |
| $\rightarrow$ KNN | 0.8218 (-2.90%) | 0.8178 (-2.71%) |
| $\rightarrow$ SVM | 0.8456 (-0.52%) | 0.8307 (-1.42%) |

**Proof:** For each sample and FreqRes module, the iteration includes an $O(S^2)$ operation for updating hidden states at each step, culminating in $O(S^2 L)$ operations for the entire series. Subsequent feature derivation, as delineated in Equation (8), requires $S^2 L$ operations for $\mathbf{X}\mathbf{X}^T$, $O(S^3)$ operations for matrix inversion, and $O(SL + S^2)$ operations for the remaining matrix multiplications. The combined complexity thus amounts to $O(S^2 L + S^3)$. Given that each FreqRes corresponds to a cycle interval less than $\kappa$, the number of FreqRes modules is bounded by $\kappa$. Hence, the overall complexity for all samples is $O(\kappa M(S^2 L + S^3))$. $\square$

# C. Supplementary Experiments

## C.1. Impact of Sequence Length on Efficiency

For a deeper insight into the computational efficiency of the proposed SARC framework, we compare its time cost with four relatively lightweight baselines: rmESN, Hydra, Rocket, MiniRocket, and LSTNet, across datasets with varying sequence lengths. Specifically, we selected six datasets, including **Crop(46)**, **ElectricDevices(96)**, **Strawberry(235)**, **FordA(500)**, **StarLightCurves(1024)**, and **HandOutlines(2709)**, with sequence lengths noted in parentheses. From the results shown in Figure 10, SARC exhibits the highest efficiency in scenarios with sequence lengths under 100, where training the classifier takes up most time. In this case, features derived by SARC have the lowest dimensionality, thus the time spent on the classifier is shorter. However, as the sequence length increases, SARC's runtime becomes slightly higher than that of rmESN, since the iterative process of the latter is simpler. Nevertheless, SARC remains faster than MiniRocket, Hydra, and Rocket, demonstrating its superior efficiency.

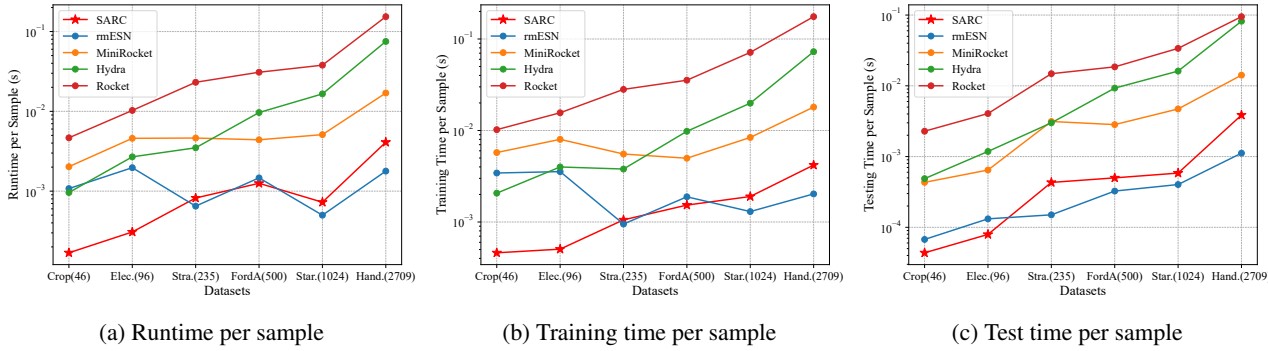

(a) Runtime per sample          (b) Training time per sample          (c) Test time per sample

*Figure 10.* Comparison of time cost across five lightweight methods on datasets with varying sequence lengths: runtime per sample in subfigure (a), training time per sample in subfigure (b), and test time per sample in subfigure (c). The time axis is on a logarithmic scale. Sequence lengths are indicated in parentheses on the x-axis.

*Table 3.* Average accuracy ranks of different methods on the 128 UCR datasets, grouped by sequence length. Bold indicates the best rank, while underlined indicates the second-best.

| Methods | 1-200 | 201-500 | 501-1000 | 1000+ |
|---|---|---|---|---|
| #Datasets | 41 | 45 | 20 | 22 |
| rmESN | 6.6098 | 7.3111 | 5.3750 | 6.9773 |
| ConvMESN | 5.5366 | 6.1222 | 5.6500 | 5.8636 |
| TimesNet | 8.1707 | 8.5667 | 8.0250 | 8.8182 |
| InceptionTime | 4.5610 | 4.1667 | 4.8250 | 4.9091 |
| COTE | **3.6463** | 4.1333 | 4.7250 | **2.2955** |
| Hydra | 4.4878 | 4.0333 | 4.5500 | 4.0909 |
| Rocket | 4.0000 | **3.3000** | 4.2000 | 4.0909 |
| MiniRocket | 4.3171 | 3.9556 | 5.1750 | 4.5909 |
| SARC | 3.6707 | 3.4111 | **2.4750** | 3.3636 |

## C.2. Accuracy Comparison by Sequence Length

In Table 3, we group 128 datasets by sequence length and report the average accuracy rank of different methods across each group. The results show that SARC achieves the best average rank of 2.475 in the length range of 501-1000. In contrast, Rocket performs best in the 201-500 range, while COTE ranks first in the remaining two groups. Despite this, SARC attains at least the second-best performance across all cases, demonstrating consistent competitiveness.

## C.3. Impact of Spectral Insights across Different Models

Table 4 provides a pairwise comparison of whether Spectral Insights (SI) are incorporated across various base models (in terms of accuracy). Specifically, Column 4 reports the win/tie/loss counts of "w/ SI" version against "w/o SI", and the p-values between them are from Wilcoxon tests. Results demonstrate that incorporating SI achieves average accuracy improvements ranging from 13.86% to 18.97% across all base models. Statistics further confirm the significance of these improvements. This validates the consistent effectiveness of our approach across a broader range of models.

*Table 4.* Pairwise accuracy comparison with and without Spectral Insights across different models. "Wins/Ties/Losses" indicate the respective counts for the model with SI versus without SI.

| Models | Avg. Acc. w/o SI | Avg. Acc. w/ SI | Wins/Ties/Losses | P-value |
|---|---|---|---|---|
| ESN | 0.6456 | 0.8353 | 122/1/5 | 6.44E-22 |
| LeakyESN | 0.7030 | 0.8416 | 117/0/11 | 3.79E-20 |
| BiESN | 0.7055 | 0.8508 | 119/3/6 | 1.58E-21 |
| DeepESN | 0.6752 | 0.8394 | 117/2/9 | 1.05E-20 |
| LSTM | 0.6212 | 0.7896 | 115/0/13 | 5.05E-20 |
| GRU | 0.6295 | 0.7926 | 114/0/14 | 2.18E-20 |

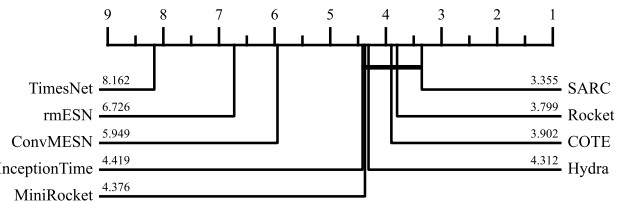

*Figure 11.* Critical Difference diagrams for different methods on 128 datasets in terms of F1-score.

## D. Extension to Multivariate Case

The proposed SARC could be adapted to multivariate scenarios in a straightforward manner: it extracts required frequencies from the mean series across variables, then applies FreqRes modules to iterate on the raw multivariate series. Another feasible extension involves separately extracting frequencies and modeling patterns for each individual variable before concatenating the derived features. While preserving variable-specific information, it would require more sophisticated implementations to ensure parallelization efficiency. Moreover, inter-variable interactions are unaddressed in this design. Further improvements could focus on the modeling of inter-variable relationships during feature extraction.

| Dataset | rmESN | Conv. | Times. | Incep. | COTE | Hydra | Rocket | Mini. | SARC |
|---|---|---|---|---|---|---|---|---|---|
| ACSF1 | 0.8300 | 0.8320 | 0.8200 | 0.8300 | **0.9360** | 0.8800 | 0.8700 | 0.8920 | 0.8500 |
| Adiac | 0.6859 | 0.7381 | 0.6609 | 0.7673 | 0.8128 | 0.8107 | 0.7754 | **0.8143** | 0.7683 |
| AllGestureWiimoteX | 0.4686 | 0.6714 | 0.4274 | 0.7600 | 0.6680 | 0.6926 | **0.7771** | 0.6951 | 0.7234 |
| AllGestureWiimoteY | 0.5826 | 0.6803 | 0.4997 | **0.7783** | 0.6831 | 0.7431 | 0.7549 | 0.7043 | 0.7583 |
| AllGestureWiimoteZ | 0.6151 | 0.6060 | 0.4323 | **0.7797** | 0.6520 | 0.6966 | 0.7494 | 0.6809 | 0.7474 |
| ArrowHead | 0.8000 | 0.8400 | 0.7943 | 0.8480 | 0.8446 | 0.7943 | 0.8011 | 0.8091 | **0.8537** |
| Beef | 0.7667 | 0.8667 | 0.5667 | 0.6600 | 0.8867 | 0.8200 | 0.8000 | 0.8667 | **0.9133** |
| BeetleFly | 0.8500 | 0.9000 | 0.6500 | 0.8500 | **0.9500** | **0.9500** | 0.9000 | 0.9100 | **0.9500** |
| BirdChicken | 0.8000 | 0.9000 | 0.8500 | 0.8000 | 0.8900 | 0.9000 | 0.9000 | 0.8000 | **0.9400** |
| BME | 0.9067 | 0.9933 | 0.8280 | 0.9933 | 0.9000 | **1.0000** | **1.0000** | **1.0000** | **1.0000** |
| Car | 0.7667 | 0.8833 | 0.6500 | 0.9000 | 0.8833 | **0.9267** | 0.8833 | 0.9167 | **0.9267** |
| CBF | 0.9016 | 0.9727 | 0.9476 | 0.9956 | 0.9920 | 0.9778 | **0.9989** | **0.9989** | 0.9882 |
| Chinatown | 0.9708 | **0.9837** | 0.9598 | 0.9825 | 0.9825 | 0.9755 | 0.9796 | 0.9825 | 0.9808 |
| ChlorineConcentration | 0.6534 | 0.7905 | 0.6766 | 0.8616 | 0.7424 | 0.7540 | 0.8061 | 0.7395 | **0.8672** |
| CinCECGTorso | 0.8061 | 0.8967 | 0.5817 | 0.8239 | **0.9981** | 0.9959 | 0.8275 | 0.8652 | 0.9468 |
| Coffee | **1.0000** | **1.0000** | **1.0000** | **1.0000** | **1.0000** | **1.0000** | **1.0000** | **1.0000** | **1.0000** |
| Computers | 0.7880 | 0.7488 | 0.6328 | 0.7880 | 0.7320 | 0.7160 | 0.7720 | 0.7880 | **0.7936** |
| CricketX | 0.5708 | 0.7046 | 0.4626 | 0.7451 | 0.8046 | 0.7923 | **0.8164** | 0.8010 | 0.8036 |
| CricketY | 0.5759 | 0.7097 | 0.4492 | 0.8231 | 0.8364 | 0.8195 | **0.8421** | 0.8313 | 0.7887 |
| CricketZ | 0.5785 | 0.7077 | 0.5005 | 0.8354 | 0.8067 | 0.8026 | **0.8467** | 0.8046 | 0.8097 |
| Crop | 0.5739 | 0.7528 | 0.6107 | **0.7608** | 0.7551 | 0.7205 | 0.7543 | 0.7427 | 0.7238 |
| DiatomSizeReduction | 0.9510 | 0.9549 | 0.9281 | 0.9837 | 0.8856 | 0.9301 | 0.9673 | 0.9190 | **0.9843** |
| DistalPhalanxOutlineAgeGroup | 0.7482 | **0.7813** | 0.7281 | 0.7511 | 0.7626 | 0.7525 | 0.7410 | 0.7237 | 0.7612 |
| DistalPhalanxOutlineCorrect | 0.7681 | **0.7899** | 0.7326 | 0.7746 | 0.7855 | 0.7739 | 0.7645 | 0.7739 | 0.7746 |
| DistalPhalanxTW | 0.7079 | 0.7079 | 0.6763 | 0.6978 | 0.7036 | 0.7022 | 0.6777 | 0.6863 | **0.7396** |
| DodgerLoopDay | 0.4375 | 0.5750 | 0.3750 | 0.5175 | 0.5500 | 0.4375 | **0.5925** | 0.5900 | 0.5875 |
| DodgerLoopGame | 0.7971 | 0.8130 | 0.4942 | 0.8551 | 0.9188 | 0.8333 | 0.8406 | 0.8725 | **0.9304** |
| DodgerLoopWeekend | 0.8768 | 0.9072 | 0.7333 | 0.9565 | 0.9710 | 0.9594 | 0.9710 | 0.9725 | **0.9826** |
| Earthquakes | 0.7597 | **0.7770** | 0.7482 | 0.7410 | 0.7482 | 0.7295 | 0.7424 | 0.7367 | 0.7568 |
| ECG200 | 0.8900 | 0.9100 | 0.7900 | 0.9000 | 0.8800 | 0.8660 | 0.9100 | **0.9140** | 0.8980 |
| ECG5000 | 0.9292 | 0.9412 | 0.9344 | 0.9398 | 0.9471 | **0.9472** | 0.9460 | 0.9437 | 0.9456 |
| ECGFiveDays | 0.9895 | 0.9965 | 0.5626 | **1.0000** | **1.0000** | 0.9981 | **1.0000** | **1.0000** | **1.0000** |
| ElectricDevices | 0.7189 | 0.7480 | 0.6782 | 0.7189 | **0.7622** | 0.7325 | 0.7268 | 0.7388 | 0.7387 |
| EOGHorizontalSignal | 0.5171 | 0.5945 | 0.4066 | 0.5663 | **0.6343** | 0.5713 | 0.6177 | 0.5707 | 0.6155 |
| EOGVerticalSignal | 0.3956 | **0.5459** | 0.3436 | 0.4762 | 0.5260 | 0.4718 | 0.5365 | 0.5110 | 0.5271 |
| EthanolLevel | 0.4736 | 0.6420 | 0.3028 | **0.7700** | 0.7076 | 0.5668 | 0.5868 | 0.5728 | 0.6036 |
| FaceAll | 0.8214 | 0.7937 | 0.6112 | 0.8130 | 0.7934 | 0.8155 | **0.9462** | 0.8046 | 0.9140 |
| FaceFour | 0.6500 | 0.9227 | 0.5114 | 0.9659 | **1.0000** | 0.8932 | 0.9773 | 0.9773 | 0.9341 |
| FacesUCR | 0.7999 | 0.9088 | 0.5637 | **0.9634** | 0.9565 | 0.9560 | 0.9611 | 0.9546 | 0.9410 |
| FiftyWords | 0.4677 | 0.7833 | 0.4958 | 0.8044 | 0.7820 | **0.8325** | 0.8290 | 0.8316 | 0.7776 |
| Fish | 0.8297 | 0.9097 | 0.8069 | 0.9851 | **0.9874** | 0.9851 | 0.9829 | 0.9623 | 0.9646 |
| FordA | 0.9117 | 0.9427 | 0.7211 | 0.9524 | 0.9465 | **0.9591** | 0.9353 | 0.9464 | 0.9561 |
| FordB | 0.7301 | 0.8032 | 0.6415 | **0.8432** | 0.8331 | 0.8175 | 0.7993 | 0.8123 | 0.8398 |
| FreezerRegularTrain | 0.9790 | 0.9920 | 0.9830 | 0.9959 | **0.9979** | 0.9949 | 0.9973 | 0.9975 | 0.9959 |
| FreezerSmallTrain | 0.9204 | 0.8575 | 0.7862 | 0.8406 | **0.9945** | 0.8640 | 0.9478 | 0.9648 | 0.9254 |
| Fungi | 0.9677 | 0.9366 | 0.8817 | **1.0000** | **1.0000** | **1.0000** | **1.0000** | **1.0000** | **1.0000** |
| GestureMidAirD1 | 0.4615 | 0.6769 | 0.5154 | 0.7385 | 0.6154 | 0.7585 | 0.7231 | 0.6938 | **0.7600** |
| GestureMidAirD2 | 0.6308 | 0.6538 | 0.6000 | 0.6846 | 0.6154 | 0.6585 | 0.6692 | 0.6338 | **0.6877** |
| GestureMidAirD3 | 0.3477 | 0.3862 | 0.2154 | 0.3923 | 0.4077 | **0.4831** | 0.3938 | 0.3800 | 0.4615 |
| GesturePebbleZ1 | 0.7047 | 0.8791 | 0.5872 | **0.9291** | 0.9070 | 0.9012 | 0.9070 | 0.9093 | 0.8837 |
| GesturePebbleZ2 | 0.6342 | 0.8241 | 0.6203 | 0.8633 | **0.8861** | 0.8696 | 0.8241 | 0.8190 | 0.8266 |
| GunPoint | 0.9533 | 0.9867 | 0.7813 | 0.9933 | **1.0000** | **1.0000** | **1.0000** | 0.9960 | **1.0000** |
| GunPointAgeSpan | 0.9557 | 0.9842 | 0.9304 | 0.9937 | 0.9968 | **1.0000** | 0.9968 | 0.9949 | 0.9956 |
| GunPointMaleVersusFemale | 0.9937 | 0.9968 | 0.9918 | 0.9968 | 0.9968 | 0.9968 | 0.9968 | 0.9968 | **1.0000** |
| GunPointOldVersusYoung | 0.9905 | 0.9346 | 0.9752 | 0.9937 | 0.9968 | 0.9924 | 0.9879 | 0.9937 | **0.9981** |
| Ham | 0.6667 | 0.6876 | 0.6095 | **0.7810** | 0.6743 | 0.7067 | 0.7143 | 0.7086 | 0.7505 |
| HandOutlines | 0.8141 | 0.9351 | 0.8357 | **0.9508** | 0.9324 | 0.9416 | 0.9405 | 0.9427 | 0.9470 |
| Haptics | 0.4714 | 0.5266 | 0.4169 | 0.5519 | 0.5675 | 0.5078 | 0.5266 | 0.5500 | **0.5773** |
| Herring | 0.6250 | 0.6281 | 0.5938 | 0.7031 | 0.6344 | **0.7313** | 0.5938 | 0.6594 | 0.6188 |
| HouseTwenty | 0.9664 | 0.8756 | 0.7983 | 0.9496 | **0.9832** | 0.9580 | 0.9580 | 0.9445 | 0.9630 |
| InlineSkate | 0.4113 | 0.4658 | 0.2658 | 0.4818 | 0.4796 | 0.4873 | 0.4662 | 0.4345 | **0.5044** |

Table 5: Summary of Accuracy Results Across 128 Datasets.

| Dataset | rmESN | Conv. | Times. | Incep. | COTE | Hydra | Rocket | Mini. | SARC |
|---|---|---|---|---|---|---|---|---|---|
| InsectEPGRegularTrain | **1.0000** | **1.0000** | 0.9960 | **1.0000** | 0.9992 | **1.0000** | **1.0000** | 0.9952 | **1.0000** |
| InsectEPGSmallTrain | 0.9197 | 0.9598 | 0.9044 | 0.8594 | 0.9671 | 0.9614 | 0.9799 | 0.9438 | **0.9831** |
| InsectWingbeatSound | 0.3904 | 0.6563 | 0.4276 | 0.6157 | 0.6553 | 0.6503 | 0.6599 | **0.6663** | 0.6542 |
| ItalyPowerDemand | 0.9592 | 0.9679 | 0.9372 | 0.9673 | 0.9670 | 0.9656 | **0.9689** | 0.9668 | 0.9668 |
| LargeKitchenAppliances | 0.8693 | 0.7520 | 0.6912 | 0.8805 | 0.7573 | 0.8635 | **0.8965** | 0.8491 | 0.8901 |
| Lightning2 | 0.7869 | 0.7705 | 0.7377 | **0.8197** | 0.7639 | 0.7377 | 0.7738 | 0.7443 | **0.8197** |
| Lightning7 | 0.7397 | 0.7671 | 0.7123 | 0.7452 | 0.7753 | 0.8082 | 0.6849 | **0.8274** |
| Mallat | 0.9494 | 0.9441 | 0.9034 | 0.8936 | **0.9719** | 0.9485 | 0.9557 | 0.9357 | 0.9597 |
| Meat | 0.9500 | 0.9367 | 0.9167 | 0.9300 | 0.9000 | 0.8933 | 0.9500 | 0.9667 | **0.9800** |
| MedicalImages | 0.7961 | 0.7924 | 0.7221 | 0.7950 | 0.7587 | 0.7553 | 0.7911 | **0.8053** | 0.7511 |
| MelbournePedestrian | 0.8731 | 0.8891 | 0.8304 | **0.9117** | 0.8921 | 0.8804 | 0.9031 | 0.8891 | 0.8815 |
| MiddlePhalanxOutlineAgeGroup | 0.5455 | 0.5338 | 0.6234 | 0.5519 | 0.5740 | 0.5909 | 0.5532 | 0.5468 | **0.6273** |
| MiddlePhalanxOutlineCorrect | 0.8213 | 0.8014 | 0.7567 | 0.8275 | **0.8488** | 0.8275 | 0.8289 | 0.8419 | 0.8474 |
| MiddlePhalanxTW | 0.5831 | 0.5519 | **0.5974** | 0.5065 | 0.5818 | 0.5299 | 0.5403 | 0.5338 | 0.5805 |
| MixedShapesRegularTrain | 0.9089 | 0.9264 | 0.8268 | 0.9563 | 0.9742 | **0.9801** | 0.9676 | 0.9720 | 0.9494 |
| MixedShapesSmallTrain | 0.8829 | 0.9037 | 0.7238 | 0.9039 | 0.9533 | **0.9620** | 0.9347 | 0.9492 | 0.8934 |
| MoteStrain | 0.6589 | 0.8922 | 0.7059 | 0.8778 | **0.9593** | 0.9462 | 0.9152 | 0.9321 | 0.8997 |
| NonInvasiveFetalECGThorax1 | 0.9077 | 0.9297 | 0.8031 | 0.9376 | 0.9296 | 0.9440 | 0.9524 | 0.9464 | **0.9539** |
| NonInvasiveFetalECGThorax2 | 0.9223 | 0.9484 | 0.8494 | 0.9532 | 0.9469 | 0.9576 | **0.9656** | 0.9651 | 0.9595 |
| OliveOil | 0.8333 | 0.7400 | 0.8333 | 0.8333 | 0.8933 | 0.9333 | 0.9000 | 0.9333 | **0.9600** |
| OSULeaf | 0.8438 | 0.8372 | 0.4463 | 0.9215 | **0.9942** | 0.9876 | 0.9347 | 0.9603 | 0.8248 |
| PhalangesOutlinesCorrect | 0.7858 | 0.7932 | 0.7744 | **0.8399** | 0.8317 | 0.8189 | 0.8275 | 0.8310 | 0.8235 |
| Phoneme | 0.3189 | 0.3315 | 0.1800 | 0.3204 | **0.3783** | 0.3205 | 0.2759 | 0.2897 | 0.3387 |
| PickupGestureWiimoteZ | 0.7600 | 0.7600 | 0.5400 | 0.6160 | 0.7160 | 0.8280 | **0.8400** | 0.6800 | 0.8160 |
| PigAirwayPressure | 0.2308 | 0.1779 | 0.1356 | 0.5529 | 0.7788 | 0.7519 | 0.3913 | **0.8702** | 0.3750 |
| PigArtPressure | 0.8712 | 0.8577 | 0.1779 | **0.9952** | 0.9856 | 0.9856 | 0.9519 | 0.9808 | 0.9837 |
| PigCVP | 0.7462 | 0.8817 | 0.1260 | 0.9173 | **0.9663** | 0.9423 | 0.9327 | 0.9096 | 0.9029 |
| PLAID | 0.8719 | 0.7568 | 0.7654 | **0.9128** | 0.8466 | 0.8648 | 0.8819 | 0.8879 | 0.9084 |
| Plane | **1.0000** | 0.9905 | 0.9714 | **1.0000** | **1.0000** | **1.0000** | **1.0000** | **1.0000** | **1.0000** |
| PowerCons | 0.9444 | 0.9467 | 0.8733 | 0.9444 | 0.9589 | 0.9511 | 0.9333 | **0.9733** | 0.9467 |
| ProximalPhalanxOutlineAgeGroup | **0.8654** | 0.8439 | 0.8449 | 0.8107 | 0.8439 | 0.8537 | 0.8449 | 0.8459 | 0.8634 |
| ProximalPhalanxOutlineCorrect | 0.8729 | 0.8873 | 0.8351 | **0.9107** | 0.8790 | 0.8942 | 0.9003 | 0.8880 | 0.9003 |
| ProximalPhalanxTW | 0.8000 | **0.8195** | 0.7912 | 0.7366 | **0.8195** | 0.8039 | 0.7912 | 0.7922 | 0.8117 |
| RefrigerationDevices | 0.5557 | 0.5253 | 0.5637 | 0.5387 | 0.5403 | 0.5013 | 0.5141 | 0.5152 | **0.5749** |
| Rock | 0.8200 | 0.8440 | 0.5000 | 0.8200 | **0.9720** | 0.8600 | 0.9000 | 0.8400 | 0.8560 |
| ScreenType | 0.5691 | 0.5104 | 0.4517 | 0.5653 | 0.4800 | 0.4736 | 0.4677 | 0.5035 | **0.5883** |
| SemgHandGenderCh2 | 0.8540 | 0.8763 | 0.7523 | 0.8687 | **0.9617** | 0.8587 | 0.9223 | 0.8870 | 0.8967 |
| SemgHandMovementCh2 | 0.5680 | 0.5680 | 0.4280 | 0.4813 | **0.7840** | 0.4609 | 0.6298 | 0.5591 | 0.6036 |
| SemgHandSubjectCh2 | 0.7698 | 0.7564 | 0.6938 | 0.7844 | **0.9356** | 0.8009 | 0.8676 | 0.8689 | 0.8884 |
| ShakeGestureWiimoteZ | 0.8400 | 0.7840 | 0.7600 | 0.8000 | 0.7720 | 0.9000 | 0.9000 | 0.8840 | **0.9040** |
| ShapeletSim | 0.9722 | 0.8144 | 0.6333 | 0.9889 | **1.0000** | 0.9833 | **1.0000** | **1.0000** | **1.0000** |
| ShapesAll | 0.6930 | 0.8120 | 0.6457 | 0.9300 | **0.9380** | 0.9273 | 0.9033 | 0.9203 | 0.8713 |
| SmallKitchenAppliances | 0.7707 | 0.7424 | 0.7205 | 0.7611 | **0.8133** | 0.8037 | 0.8091 | 0.8011 | 0.7675 |
| SmoothSubspace | 0.7733 | 0.9000 | 0.8200 | 0.9667 | **0.9773** | 0.9333 | **0.9773** | 0.9507 | 0.8880 |
| SonyAIBORobotSurface1 | 0.9408 | 0.8812 | 0.8106 | 0.9334 | 0.8439 | 0.9185 | 0.9158 | 0.8892 | **0.9464** |
| SonyAIBORobotSurface2 | 0.9412 | 0.8959 | 0.7142 | 0.9234 | 0.9383 | **0.9467** | 0.9196 | 0.9037 | 0.9324 |
| StarLightCurves | 0.9589 | 0.9189 | 0.9292 | 0.9777 | 0.9795 | **0.9814** | 0.9806 | 0.9812 | 0.9813 |
| Strawberry | 0.9514 | **0.9827** | 0.9200 | 0.9768 | 0.9751 | 0.9719 | 0.9811 | 0.9789 | 0.9757 |
| SwedishLeaf | 0.9347 | 0.9274 | 0.8531 | 0.9616 | 0.9507 | **0.9798** | 0.9616 | 0.9648 | 0.9533 |
| Symbols | 0.8400 | 0.9254 | 0.8868 | 0.9739 | 0.9727 | **0.9791** | 0.9729 | 0.9783 | 0.9574 |
| SyntheticControl | 0.7033 | 0.9967 | 0.9800 | 0.9980 | **1.0000** | 0.9767 | **1.0000** | 0.9920 | 0.9993 |
| ToeSegmentation1 | 0.9211 | 0.8526 | 0.6754 | 0.9693 | **0.9737** | 0.9632 | 0.9518 | 0.9658 | 0.9518 |
| ToeSegmentation2 | 0.9077 | 0.9154 | 0.8154 | 0.9308 | **0.9538** | 0.9108 | 0.9231 | 0.9077 | 0.9185 |
| Trace | **1.0000** | 0.9800 | 0.8900 | **1.0000** | **1.0000** | **1.0000** | **1.0000** | **1.0000** | **1.0000** |
| TwoLeadECG | **0.9991** | 0.9473 | 0.7034 | 0.9965 | 0.9939 | **0.9991** | **0.9991** | 0.9982 | **0.9991** |
| TwoPatterns | 0.5100 | 0.9958 | 0.8008 | **1.0000** | 0.9975 | 0.9858 | **1.0000** | 0.9961 | 0.9961 |
| UMD | 0.8403 | 0.9722 | 0.9861 | 0.9653 | 0.9931 | 0.9889 | 0.9931 | 0.9931 | **0.9944** |
| UWaveGestureLibraryAll | 0.7085 | 0.9515 | 0.6289 | 0.9370 | 0.9674 | 0.9706 | **0.9739** | 0.9691 | 0.9523 |
| UWaveGestureLibraryX | 0.6602 | 0.7286 | 0.6640 | 0.8110 | 0.8130 | **0.8575** | 0.8557 | 0.8510 | 0.8211 |
| UWaveGestureLibraryY | 0.6884 | 0.6304 | 0.6266 | 0.7499 | 0.7367 | **0.7931** | 0.7772 | 0.7740 | 0.7452 |
| UWaveGestureLibraryZ | 0.6912 | 0.6940 | 0.6116 | 0.7552 | 0.7633 | **0.7992** | 0.7950 | 0.7968 | 0.7697 |

Table 5: Summary of Accuracy Results Across 128 Datasets.

| Dataset | rmESN | Conv. | Times. | Incep. | COTE | Hydra | Rocket | Mini. | SARC |
|---|---|---|---|---|---|---|---|---|---|
| Wafer | 0.9904 | 0.9972 | 0.9883 | 0.9978 | 0.9997 | **1.0000** | 0.9979 | 0.9993 | 0.9978 |
| Wine | 0.7778 | 0.7444 | 0.8148 | 0.7222 | 0.8481 | **0.8815** | 0.7815 | 0.8667 | 0.8667 |
| WordSynonyms | 0.5185 | 0.6890 | 0.4875 | 0.7132 | 0.6994 | 0.7382 | **0.7467** | 0.7251 | 0.7213 |
| Worms | 0.7403 | 0.6909 | 0.5974 | 0.7792 | 0.7117 | 0.7325 | 0.7273 | 0.6831 | **0.7896** |
| WormsTwoClass | 0.8052 | 0.7662 | 0.6234 | 0.7792 | 0.8182 | 0.7610 | 0.8052 | 0.7532 | **0.8286** |
| Yoga | 0.8451 | 0.8591 | 0.7899 | 0.9050 | 0.9203 | **0.9210** | 0.9066 | 0.9137 | 0.8793 |

Table 5: Summary of Accuracy Results Across 128 Datasets.

