# OpenReview forum: "Spectral-Aware Reservoir Computing for Fast and Accurate Time Series Classification"
_ICML.cc/2025/Conference — ICML 2025 poster_

### Official Review · Reviewer_CNcc · 2025-03-12

**Overall Recommendation:** 4

**Summary:**

This paper introduces a novel method for time series classification. The approach begins by decomposing the time series into multiple prominent frequencies, each representing different cyclic patterns in the data. For each extracted frequency, a reservoir computing model—called FreqRes—is applied to generate complex hidden states. These hidden states are then used to produce individual readouts, which are subsequently concatenated to form a final representation for classification. Experiments on the UCR dataset demonstrate that this method is effective and outperforms competing approaches.

**Claims And Evidence:**

**Key Claims and Comments**

- *Incorporating Spectral Information into Echo State Neural Networks:*
Ablation studies in Section 5.3 (particularly Figure 8a) demonstrate that integrating spectral insights into BiESN enhances performance across most datasets, providing strong evidence of its effectiveness. However, it would be valuable to see similar analyses for other architectures, such as ENS, LeakyESN, DeepESN, and LSTM/GRU, to determine whether spectral insights benefit all models.

- *Computational Efficiency and Performance:*
Figure 5 presents a computational cost analysis, highlighting the method’s low computational overhead alongside a time complexity evaluation. Additionally, comparisons of F1-score and accuracy against multiple competitors show a performance improvement, reinforcing the effectiveness of the approach.

**Essential References Not Discussed:**

No essential references that I know.

**Experimental Designs Or Analyses:**

The experimental design is well-executed, with a comprehensive set of benchmarks (128 UCR datasets) used for comparison. The chosen metrics are appropriate, and p-values are computed to assess statistical significance. However, the authors should place greater emphasis on evaluating the impact of spectral insights—the main contribution of the paper. Specifically, they should demonstrate that these insights not only enhance performance beyond BiESN but also apply to other model variants.

**Methods And Evaluation Criteria:**

*Evaluation Criteria:*
The authors evaluate their method based on computational cost and several widely accepted metrics in the community, including F1-score, accuracy, Win/Tie/Loss analysis, and statistical significance testing using p-values. Additionally, the benchmarks used are standard, ensuring a fair comparison. However, a limitation is that the method only applies to univariate time series, whereas the main challenges often lie in the multivariate setting.

*Proposed Method:*
Incorporating spectral insights into time series analysis is a logical approach, as frequency-based representations are a well-established and widely used method for capturing time series patterns.

**Other Comments Or Suggestions:**

1- Need for incorporating the baselines of using classifiers directly on the concatenation of raw extracted from each frequency.
2 Need to comment on multivariate extensions


**After rebutal**
I increase my score to 4 given the answer of the authors

**Other Strengths And Weaknesses:**

**Strengths:**
- Extensive experimental studies
- Well-written article with a clear motivation

**Weaknesses:**
- The method is limited to univariate time series classification.
- A direct comparison should be made between applying classifiers (e.g., Ridge, Random Forest) directly to the extracted frequency components and using reservoir computing. This would clarify whether the reservoir computing truly adds value, especially given that the reservoir size does not appear to be a key factor. If the reservoir computing is not significantly improving performance, it may be unnecessary, as a direct classification approach could be more computationally efficient.

**Questions For Authors:**

1- Could the authors explain at least experimentally the need for the reservoir computing?
2- Can the authors discuss the extension to multivariate time series classification?
3- Could the authors make some experiments regarding the number of frequencies extracted ?

**Relation To Broader Scientific Literature:**

To the best of my knowledge, this is the first work to incorporate spectral insights into Echo State Neural Networks (ESNs), despite the existence of several studies leveraging frequency information for classification. In the specific context of ESNs, this approach appears to be the first in this direction, making it the key innovation of the paper

**Theoretical Claims:**

1- No theoretical claims in the paper.

---

> ### Author Rebuttal · Authors · 2025-03-28
>
> We express our gratitude to the reviewer for your time and recognition. Detailed responses to the raised concerns are listed below. We agree with the suggestions mostly and will clarify those details.
>
> **1. Impact of Spectral Insights across Different Models**
>
> The table below provides a comparison of whether Spectral Insights (SI) are incorporated across various base models (in terms of accuracy). Specifically, Row 4 reports the win/tie/loss counts of "w/ SI" version against "w/o SI", and the p-values between them are from Wilcoxon tests.
>
> Results demonstrate that incorporating SI achieves average accuracy improvements ranging from 13.86\% to 18.97\% across all models. Statistics further confirm the significance of these improvements. This validates the consistent effectiveness of our approach across a broader range of models.
>
> ---
> |   | ESN | LeakyESN | BiESN | DeepESN | LSTM | GRU |
> |--|--|--|--|--|--|--|
> | Avg. Acc. w/o SI | 0.6456 | 0.7030 | 0.7055 | 0.6752 | 0.6212 | 0.6295 |
> | Avg. Acc. w/ SI | 0.8353 | 0.8416 | 0.8508 | 0.8394 | 0.7896 | 0.7926 |
> | Wins/Ties/Losses | 122/1/5 | 117/0/11 | 119/3/6 | 117/2/9 | 115/0/13 | 114/0/14 |
> | P-value | 6.44E-22 | 3.79E-20 | 1.58E-21 | 1.05E-20 | 5.05E-20 | 2.18E-20 |
> ---
>
> **2. The Need for Reservoir Computing**
>
> As Eq. (6) shows, the FreqRes modules iterate on raw time series and implicitly utilize frequency information. Thus, an explicit frequency component for direct classification could be unavailable. In response to the reviewer's expectation, the following features in our experiments enable comparison and demonstrate the contribution of reservoir computing (RC):
>
> - $O$: Original time series data
> - $F_1$: For each $f \in \mathcal{F}$ and $p = [L/f]$, compute a differenced series of the raw sequence and concatenate statistical features, including mean absolute value, standard deviation, kurtosis, zero-crossing rate, and first-order autocorrelation coefficient.
> - $F_2$: For each $f \in \mathcal{F}$, extract frequency band components (centered at $f$, 10% relative bandwidth) and concatenate features, including relative band energy ratio alongside the above five statistical features.
> - $F_3$: Feature vectors composed of FFT amplitude values at each $f \in \mathcal{F}$.
>
> The average accuracies using Ridge and Random Forest classifiers with these features are presented below. This comparison highlights the necessity of RC and FreqRes modules, as they achieve significantly better performance by capturing temporal dependencies at multi-scale frequencies.
>
> ---
> |   | $O$ | $F_1$ | $F_2$ | $F_3$ | $O+F_1$ | $O+F_2$ | $O+F_3$ | $O+F_{1,2,3}$ | SARC |
> |--|--|--|--|--|--|--|--|--|--|
> | Ridge | 0.6475 | 0.7112 | 0.6322 | 0.6178 | 0.7270 | 0.6961 | 0.7048 | 0.7422 | 0.8508 |
> | RF | 0.7209 | 0.7281 | 0.6920 | 0.6610 | 0.7712 | 0.7680 | 0.7430 | 0.7894 | 0.8391|
> ---
>
>
> **3. Extension to Multivariate TSC**
>
> The proposed SARC is currently adapted to multivariate scenarios in a straightforward manner: it extracts required frequencies from the mean series across variables, then applies FreqRes modules to iterate on the raw multivariate series. This approach proves effective, as evidenced by the submitted supplementary materials, which contain a demo achieving ~98% accuracy on the "ArticularyWordRecognition" dataset from the UEA archive.
>
> Another feasible extension involves separately extracting frequencies and modeling patterns for each individual variable before concatenating the derived features. While preserving variable-specific information, it would require more sophisticated implementations to ensure parallelization efficiency. Moreover, inter-variable interactions are unaddressed in this design. Further improvements could focus on the modeling of inter-variable relationships during feature extraction, and we plan to explore this in future studies.
>
> **4. Impact of Frequency Quantity**
>
> As described in Section 4.1.2, the extracted frequency set $\mathcal{F}$ consists of several "root" frequencies and their sub-harmonics. Figure 7(c) in the main paper has visualized the impact of threshold $\kappa$ (which controls the number of sub-harmonics) on average accuracy across 128 datasets. Here, we offer an analysis regarding the impact of "root" frequency quantity.
>
> The table below shows SARC's average accuracy when selecting and extending only the top-$\alpha$ smallest "root" frequencies. Similar to the trend with $\kappa$, SARC achieves better performance as $\alpha$ increases (i.e., more frequencies are included). Notably, the number of extracted "root" frequencies is less than 10 for most datasets, resulting in identical accuracy when $\alpha=9$ and $10$ on these datasets. This explains why the average accuracy growth gradually plateaus when $\alpha$ approaches 10.
>
> ---
> | $\alpha=$ | 1 | 2 | 3 | 4 | 5 | 6 | 7 | 8 | 9 | 10 |
> |--|--|--|--|--|--|--|--|--|--|--|
> | Avg. Acc.| 0.6863 | 0.7641 | 0.7937 | 0.8140 | 0.8279 | 0.8373 | 0.8454 | 0.8503 | 0.8507 | 0.8508 |
> ---

---

> > ### Comment · Reviewer_CNcc · 2025-04-09
> >
> > Thank you to the authors for providing a detailed response to my comments. I am satisfied with their explanations, and I find their arguments convincing in the univariate case. However, I would encourage the authors to explore the multivariate case further and explain how the extension could be implemented in the paper. Additionally, benchmarking against UEA datasets would enhance the impact of their work.

---

> > > ### Author Response · Authors · 2025-04-09
> > >
> > > We sincerely appreciate the reviewer for the acknowledgment and valuable suggestion. We will further explore the multivariate case in depth and provide a detailed explanation of its implementation in the paper. We will also make efforts to conduct experiments on UEA datasets to assess the feasibility of the extension. Thanks again for the reviewer's time and helpful comments!

---

### Official Review · Reviewer_X5Kk · 2025-03-13

**Overall Recommendation:** 3

**Summary:**

Typical reservoir computing (RC) considers recursive updates from adjacent states and has difficulty handling long-term dependencies. For this issue, this paper proposes a Spectrum-Aware Reservoir Computing framework (SARC) that incorporates spectral insights to enhance long-term dependency modeling. Prominent frequencies are first extracted to reveal explicit or implicit periodic patterns. For each prominent frequency, SARC further integrates a frequency-informed reservoir network (FreqRes) to fully capture the sequential and periodic dynamics, thereby deriving effective dynamic features. By synthesizing these features at different frequencies, SARC provides a multi-scale analysis of temporal dynamics and improves the modeling of long-term dependencies. Extensive experiments on pubic datasets demonstrate that SARC achieves state-of-the-art results.

**Claims And Evidence:**

The claim that SARC provides multiscale analysis of temporal dynamics and improves modeling of long-range dependencies is supported by a large body of experimental evidence, including experimental results on multiple datasets. However, the theoretical analysis and description of the SARC framework could be clearer, especially in how long-term dependencies are captured.

**Essential References Not Discussed:**

The paper cites key references, but would benefit from more discussion on the connection to other spectral analysis techniques in time series modeling, particularly those utilizing advanced wavelet transforms and adaptive frequency decomposition methods.

**Experimental Designs Or Analyses:**

The experiments are well designed and the benchmarks are clear. The comparison with eight state-of-the-art baselines and the use of statistical significance tests such as the Wilcoxon Signed-Rank Test strengthen the validity of the results.

**Methods And Evaluation Criteria:**

The proposed SARC framework is suitable for handling long-term dependencies in RC. Besides, the comparison of SARC with existing RC methods helps confirm the effectiveness of SARC.

**Other Comments Or Suggestions:**

1. Provide a clearer definition of the frequency-informed reservoir network (FreqRes), especially its theoretical properties, stability, and impact on modeling long-term dependencies.
2. Some hyperparameter choices (e.g., spectral radius, input scaling) appear to be empirically driven. A more systematic sensitivity analysis would enhance experimental validity.

**Other Strengths And Weaknesses:**

Strengths:
1. This paper conducts comprehensive experiments on 128 UCR datasets to demonstrate the effectiveness and efficiency of the proposed SARC framework. Comparisons with multiple state-of-the-art methods further strengthen its empirical contribution.
2. This paper clearly points out the limitations of existing reservoir computing methods in dealing with long-term dependencies and periodic patterns. The introduction of spectrum-aware modeling makes a meaningful contribution to time series classification.
Weaknesses:
1. Although this paper introduces the FreqRes component to enhance long-term dependency modeling, its mathematical formulation lacks sufficient clarity, especially in terms of theoretical guarantees.
2. This paper does not fully compare its spectrum-aware approach with other frequency-domain methods. A more in-depth discussion will help position SARC in the broader field of spectrum-based machine learning techniques.

**Questions For Authors:**

1. Could you provide a more formal and detailed mathematical explanation of the frequency-informed reservoir network (FreqRes)? A clearer formulation would help strengthen the theoretical justification of the proposed method.
2. How does SARC compare to other frequency-aware time series models, such as adaptive Fourier-based methods or alternative wavelet-based approaches? A more detailed discussion could better position SARC within the broader landscape of spectral-based time series classification.

**Relation To Broader Scientific Literature:**

The paper positions itself within the broader context of time series classification and reservoir computing, comparing the Spectral-Aware Reservoir Computing (SARC) framework to traditional reservoir computing models and deep learning approaches. It highlights the advantages of integrating spectral analysis for improved long-term dependency modeling. However, a more comparison with other spectral-aware approaches would be appreciated.

**Theoretical Claims:**

The theoretical claims about incorporating spectral insights to enhance the modeling of long-term dependencies are well-founded. However, the mathematical formulation of the frequency-informed reservoir network (FreqRes) could be more explicitly defined, especially about the convergence properties of its cyclical connections.

---

> ### Author Rebuttal · Authors · 2025-03-31
>
> We sincerely appreciate the reviewer’s efforts and recognition. Our responses to the raised concerns are listed below:
>
> **1. Mathematical Explanation of FreqRes**
>
> We formally define a general FreqRes, and then give some theoretical claims for ESN-based FreqRes, as described in Eq. (6).
>
> **(1) Definition of FreqRes**
>
> Let the following recursive process define a general RC model:
> $$
> \begin{aligned}
> \mathbf{\hat{x}}(n) &= \mathbf{W} \mathbf{x}(n-1), \\\\
> \mathbf{x}(n) &= \sigma\left(\mathbf{\hat{x}}(n), \mathbf{u}(n)\right),
> \end{aligned}
> $$
> where $\mathbf{x}(n)$ is the hidden state; $\mathbf{u}(n)$ is the input; $ \mathbf{W}$ is the reservoir weight matrix; and $ \sigma(\cdot) $ is the activation.
>
> The corresponding FreqRes with a frequency $f$ redefines its recursive process as:
> $$
> \begin{aligned}
> \mathbf{\hat{x}}(n) &= \mathbf{W}\_{\text{s}} \mathbf{x}(n-1) + \mathbf{W}\_{\text{c}} \mathbf{x}(n-p), \\\\
> \mathbf{x}(n) &= \sigma\left(\mathbf{\hat{x}}(n), \mathbf{u}(n)\right).
> \end{aligned}
> $$
> Here, $L$ is series length ; $p = [L/f]$ is cycle length; $ \mathbf{W}\_{\text{s}}$ and $\mathbf{W}\_{\text{c}}$ are weight matrices modulating sequential and cyclical dependencies.
>
> **(2) Lipschitz Continuity Analysis**
>
> To analyze state stability, we treat the input influence $\mathbf{W}\_{\text{in}} \mathbf{u}(n)$ as fixed.
>
> **Claim 1:** The Lipschitz constant of state update function for the ESN-based FreqRes is bounded by $\sqrt{ \Vert\mathbf{W}\_{\text{s}}\Vert_2^2 + \Vert\mathbf{W}\_{\text{c}}\Vert_2^2 }$. Thus, $\Vert\mathbf{W}\_{\text{s}}\Vert_2^2 + \Vert\mathbf{W}\_{\text{c}}\Vert_2^2 < 1$ ensures a contraction mapping and guarantees state stability.
>
> Proof sketch：
>
> Noting that $\tanh$ is 1-Lipschitz and applying the Cauchy-Schwarz inequality:
> $$
> \begin{align}
> \Vert \mathbf{A} \mathbf{x}\_{1} + \mathbf{B} \mathbf{x}\_{2} - \mathbf{A} \mathbf{x}\_{1}' - \mathbf{B} \mathbf{x}\_{2}'\Vert\_{2}
> & \leq \Vert \mathbf{A}\Vert\_{2} \Vert(\mathbf{x}\_{1} - \mathbf{x}\_{1}')\Vert\_{2} + \Vert\mathbf{B}\Vert\_{2} \Vert(\mathbf{x}\_{2} - \mathbf{x}\_{2}')\Vert\_{2}\\\\
> & \leq \sqrt{\Vert \mathbf{A}\Vert\_{2}^{2} + \Vert \mathbf{B}\Vert\_{2}^{2}}\sqrt{\Vert\mathbf{x}\_{1} - \mathbf{x}\_{1}'\Vert\_{2}^{2} + \Vert\mathbf{x}\_{2} - \mathbf{x}\_{2}'\Vert\_{2}^{2})}
> \end{align}
> $$
>
> **(3) Discussion on Echo State Property**
>
> **Claim 2:** Given $p = [L/f]$, an ESN-based FreqRes with reservoir size $s$ satisfies the ESP  $\iff$ a specially structured ESN with reservoir size $ps$ does.
>
> Proof sketch：
>
> Let $\mathbf{U}(n) = [\mathbf{u}(n); \mathbf{u}(n-1); \dots; \mathbf{u}(n-p+1)]$, and $\mathbf{X}(n) = [\mathbf{x}(n); \dots; \mathbf{x}(n-p+1)]$.
>
> The state update could be rewritten as: $$\mathbf{X}(n) = \sigma\left(\begin{bmatrix}
> \mathbf{W}\_{\text{s}} & \mathbf{0} & \mathbf{W}\_{\text{c}} \\\\
> \mathbf{I}\_{s} & \mathbf{0} & \mathbf{0}\\\\
> \mathbf{0} & \mathbf{I}\_{(p-2)s} & \mathbf{0}\\\\
> \end{bmatrix}\mathbf{X}(n-1) + \begin{bmatrix}
> \mathbf{W}\_{\text{in}} &  &  \\\\
>  & \mathbf{0}\_{s} & \\\\
>  &  & \mathbf{0}\_{(p-2)s}\\\\
> \end{bmatrix}\mathbf{U}(n)\right)$$
>
> Here, $\sigma$ is 1-Lipschitz, applying $\tanh$ to the first $s$-dimensional part and identity function to the rest, which aligns with the common assumptions used in ESP discussions. Thus, existing theories can be applied to derive constraints on $\mathbf{W}\_{\text{s}}$ and $\mathbf{W}\_{\text{c}}$.
>
> **(4) Hyperparameter Analysis**
>
> Due to space constraints, we briefly discuss spectral radius (SR) and input scaling (IS) choices. Based on the above analysis, restricting SR $< 1$ is a necessary condition for state stability, justifying the range selection in Fig. 8b. For IS, Fig. 7a shows that overly small values markedly weaken input influence and degrade performance. With further validation, we find that excessively large IS also reduce performance, likely because states activated by $\tanh$ saturate at $\pm 1$, as weights and historical states remain bounded.
>
> **2. Comparison with Spectral-Based Methods**
>
> We compare SARC with Fourier or wavelet-based methods in accuracy using results from their original papers. The table below shows SARC's consistent outperformance in pairwise comparisons, highlighting its competitiveness in spectrum-based TSC.
>
> These methods include:
> - **AMSWR('21):** Learnable multi-scale wavelet decomposition via adaptive CNNs. DOI: 10.3390/info12060252
> - **SFCC('22):** Stratified frequency recombination for data augmentation to enhance ResNet. DOI: 10.1007/s11063-022-10965-9
> - **TF-Net('22):** Two-stage fusion of ResNets trained on raw series and wavelet representations. DOI: 10.1007/s10489-022-03485-5
> - **CoInception('24):** Noise-resilient wavelet contrastive views with cross-view alignment. DOI: 10.1109/ICDM59182.2024.00041
>
> ---
> | | AMSWR| SFCC|TF-Net|CoIncep.|
> |--|--|--|--|--|
> |#Dataset|85|128|85|128|
> |Avg. Acc. $\uparrow$|2.15%|1.8%|0.53%|0.82%|
> |Win/Tie/Loss|60/4/21|76/6/46|43/3/39|71/6/51|
> |P-value|1.17E-05|3.61E-03|6.01E-01|5.75E-02|
> ---

---

### Official Review · Reviewer_uL8Q · 2025-03-13

**Overall Recommendation:** 4

**Summary:**

The paper enhances reservoir computing (RC) for TSC by incorporating spectral insights. It extracts prominent frequencies to identify cyclical patterns and refines RC module to capture cyclical dynamics. Features from next-step prediction tasks are used for classification. Extensive experiments on the UCR 128 archive show state-of-the-art performance with superior efficiency, highlighting the potential of RC in time series modeling.

**Claims And Evidence:**

The paper is rigorous and main claims are well-supported:
1) Superior performance is shown by comparing with SOTAs across full UCR benchmarks.
2) High efficiency is validated via time and complexity analysis.
3) Ablation studies confirm the effectiveness of spectral insights and adaptability to various RC models.
4) Other minor design and claims are supported by reasonable explanations and experiments.

**Essential References Not Discussed:**

Essential works in both the RC and TSC fields are discussed.

**Experimental Designs Or Analyses:**

The experimental design is sound and well-structured, with rich datasets and comprehensive baselines. Diverse metric indicators allow convincing comparisons. Additionally, extensive ablation studies and hyperparameter analyses support the claims, and the provided code aids reproducibility. Only a minor clarification is needed on data preparation (see Questions).

**Methods And Evaluation Criteria:**

The proposed method and evaluation criteria are suitable for TSC. Combining spectral analysis with RC is novel and addresses long-term dependencies effectively. The refined RC module, along with specially designed features, enables fast and accurate classification. The UCR 128 archive and strong baselines like mini rocket and cote ensure solid validation.

**Other Comments Or Suggestions:**

I have no other comments.

**Other Strengths And Weaknesses:**

Besides the above, there are two notable strengths:
1) The derivation of dynamic features is interesting, using the predictive model as a sequence representation. This idea could inspire further cross-task research.
2) The writing is a strong point with high readability. Motivation is well-justified, and technical details are clear. Each step is explained with its purpose, ensuring rigor and logical flow.
The paper looks promising in many aspects, and I found no obvious weaknesses.

**Questions For Authors:**

Data preparation: Some UCR datasets have variable lengths or NaN values. They can greatly affect the performance of RC-based and recurrent models. How were they processed in the experiments?

**Relation To Broader Scientific Literature:**

RC techniques have been applied to TSC, but often with conventional structures. The paper advances RC by integrating spectral insights, aligning with trends in time series modeling, such as multi-scale and periodicity analysis. This design improves accuracy while maintaining high efficiency. Notably, the proposed method offers a generalizable extension to various RC models, representing an significant  advancement in the field.

**Theoretical Claims:**

Time complexity analysis appears correct based on appendix proofs, showing linear complexity with respect to sample size and length, supporting the efficiency claim.

---

> ### Author Rebuttal · Authors · 2025-03-28
>
> We are thankful for the reviewer’s recognition of the work. The processing of datasets with variable lengths or NaN values is clarified as follows:
>
> - For middle NaN values (i.e., real values exist on both sides of the NaN), we employ the interpolation for imputation.
> - Then, we shift real values in each time series sample to the rightmost positions of the tensor.
> - Finally, the rest leading NaNs are filled with zeros.
>
> Shifting real values to the right is critical due to the recurrent nature of FreqRes. Specifically, this ensures that the initial hidden states start as zeros and transition to normal iteration when real values appear. In addition, the leading zero states do not affect subsequent ridge regression computations.

---

### Official Review · Reviewer_wNEC · 2025-03-13

**Overall Recommendation:** 4

**Summary:**

The paper proposes SARC, a novel framework that combines spectral analysis with RNN-based models for time series classification. SARC identifies prominent frequencies and captures corresponding temporal dynamics. The framework analyzes time series on multiple scales, achieving state-of-the-art accuracy on the full UCR archive and showing exceptional speed.

**Claims And Evidence:**

Main claims of the paper, including the listed contributions, complexity analysis, and design tricks, are supported by reasonable evidence.

**Essential References Not Discussed:**

I believe the current references are sufficient. Given the paper's emphasis on efficiency, excluding some large models like RDST and HC2 is acceptable.

**Experimental Designs Or Analyses:**

The experimental design is targeted and thorough, supporting main claims and showcasing the proposed method's effectiveness. It compares several relevant reservoir computing methods and state-of-the-art approaches. The analyses are detailed, evaluating accuracy, F1-score, and time across multiple dimensions.

**Methods And Evaluation Criteria:**

The proposed method uses a novel reservoir-based RNN network to analyze cyclical patterns at multiple scales, achieving high accuracy and very fast speed for TSC. The utilized UCR datasets provide diverse characteristics, and comparisons include highly efficient MiniRocket. Statistical tests (Wilcoxon Test, CD diagrams) further enhance evaluation reliability.

**Other Comments Or Suggestions:**

None.

**Other Strengths And Weaknesses:**

Strengths:
1. The algorithm's lightweight nature is a key highlight, especially for architectures comprising multiple recurrent structures. This encourages further exploration of reservoir computing.
2. Technical details are clearly presented, making the reservoir model's properties easy to grasp.
3. The proposed dynamic features break classification into forecasting + classification, suggesting a new pathway to leveraging forecasting models for classification.
4. The architecture emphasizes accurate representation of time series, showing potential for broader applications like forecasting and imputation.

Weaknesses:
1. While described in footnotes and appendix, how the framework adapts to different RC models is not entirely intuitive.
2. Analyzing performance on varying lengths could further enhance evaluation.

Despite minor issues, I find this paper acceptable and capable of inspiring new ideas.

**Questions For Authors:**

1. Please provide formulas to further clarify SARC's iteration when based on BiESN, which is the primary model evaluated in the experiments.
2. Since high efficiency is a core contribution, how does SARC achieve such performance?

**Relation To Broader Scientific Literature:**

The paper adopts a multi-periodicity perspective to analyze time series, which may be influenced by prior work such as [1]. The difference is that it employs an ensemble of reservoir-based models to capture multi-scale temporal dynamics. This architecture is rare in the current literature and could offer new perspectives for time series modeling.
[1] Wu et al. "TimesNet: Temporal 2D-variation modeling for general time series analysis." ICLR 2023.

**Theoretical Claims:**

The theoretical claims involving complexity analysis and are proven correct.

---

> ### Author Rebuttal · Authors · 2025-03-29
>
> We appreciate the reviewer’s efforts and acknowledgement. Below, we provide detailed responses to your concerns:
>
> **1. Comparison by Sequence Length**
>
> In the table below, we group 128 datasets by sequence length and report the average rank of different methods across each group. The results show that SARC achieves the best average rank of 2.475 in the (500, 1000] case. In contrast, Rocket performs best in the (200, 500] range, while COTE ranks first in the remaining two groups. Despite this, SARC attains at least the second-best performance across all cases, demonstrating consistent competitiveness.
>
> ---
> | Length | Counts | rmESN | Conv. | Times. | Incep. | COTE | Hydra | Rocket | Mini. | SARC |
> |-----|-----|-------|----------|----------|---------------|---------|---------|--------|------------|--------|
> | (0, 200] | 41 | 6.6098 | 5.5366 | 8.1707 | 4.5610 | 3.6463 | 4.4878 | 4.0000 | 4.3171 | 3.6707 |
> | (200, 500] | 45 | 7.3111 | 6.1222 | 8.5667 | 4.1667 | 4.1333 | 4.0333 | 3.3000 | 3.9556 | 3.4111 |
> | (500, 1000] | 20 | 5.3750 | 5.6500 | 8.0250 | 4.8250 | 4.7250 | 4.5500 | 4.2000 | 5.1750 | 2.4750 |
> | (1000, 2844] | 22 | 6.9773 | 5.8636 | 8.8182 | 4.9091 | 2.2955 | 4.0909 | 4.0909 | 4.5909 | 3.3636 |
> ---
>
> **2. SARC’s Iteration Based on BiESN**
>
> SARC's iteration based on BiESN has been introduced through Equations (6)-(8) and footnote 3 of the main paper. For further clarification, we formally describe the computation of FreqRes based on BiESN.
>
> Given an input time series $\mathbf{u}$ of length $L$, a specific frequency $f$, and $p = [L/f]$, this FreqRes module iterates bidirectionally as follows:
>
> - Forward hidden state update:
> $$ \mathbf{x}^{\text{f}}(n) = \tanh \left(\mathbf{W}\_{\text{s}}^{\text{f}}\mathbf{x}^{\text{f}}(n-1) + \mathbf{W}\_{\text{c}}^{\text{f}}\mathbf{x}^{\text{f}}(n-p) + \mathbf{W}\_{\text{in}}^{\text{f}}\mathbf{u}(n) \right),$$
>
> - Backward hidden state update:
> $$\mathbf{x}^{\text{b}}(n) = \tanh \left(\mathbf{W}\_{\text{s}}^{\text{b}}\mathbf{x}^{\text{b}}(n+1) + \mathbf{W}\_{\text{c}}^{\text{b}}\mathbf{x}^{\text{b}}(n+p) + \mathbf{W}\_{\text{in}}^{\text{b}}\mathbf{u}(n) \right),$$
>
>
> Here, superscripts f/b denote forward/backward directions; $\mathbf{x}^{\text{f}}(n) = 0$ for $n \leq 0$; and $\mathbf{x}^{\text{b}}(n) = 0$ for $n > L$.
>
> After computing all hidden states, the states at the same time step are concatenated as: $\mathbf{x}(n) = \begin{bmatrix} \mathbf{x}^{\text{f}}(n) \\\\ \mathbf{x}^{\text{b}}(n) \end{bmatrix}$, which is then used to compute dynamic features as described in Equations (7) and (8).
>
> **3. Discussion on SARC's Efficiency**
>
> We attribute SARC’s high efficiency to three main factors:
>
> - Based on reservoir computing, SARC eliminates the need for iterative training and time-consuming gradient backpropagation.
> - Different FreqRes modules operate independently, enabling efficient parallel processing.
> - FreqRes maintains nearly linear complexity with respect to data size, ensuring scalability.

---

### Decision · Program_Chairs · 2025-05-01

**Decision:**

Accept (poster)

**Comment:**

The paper makes a significant contribution to time series modeling through a method that uses "spectral insights to enhance long-term dependency modeling".

The reviewers appreciated the novelty of the method, its broad applicability, the clarity of the presentation and the strength of the results. The reviewers also determined that the claims of the paper are well supported by the experiments, in particular the ablation studies.

Questions raised by the reviewers involving some of the technical details of the method were successfully addressed in the rebuttal. The authors also provided a comparison of the performance of different methods for timeseries of different lengths, an experiment motivating the need for reservoir computing, and a study on incorporating "spectral insights" into different base models.

Overall, the paper appears ready for publication.